# Design Criteria for SGD Preconditioners: Local Conditioning, Noise Floors, and Basin Stability

**Mitchell Scott**                                                          *mitchell.scott@emory.edu*
*Department of Mathematics*
*Emory University*

**Tianshi Xu**                                                                  *tianshi.xu@emory.edu*
*Department of Mathematics*
*Emory University*

**Ziyuan Tang**                                                                    *tang0389@umn.edu*
*Department of Computer Science*
*University of Minnesota Twin Cities*

**Alexandra Pichette-Emmons**                                        *hugolarochelle@google.com*
*Department of Mathematics*
*University of Kentucky*

**Qiang Ye**                                                                            *qye3@uky.edu*
*Department of Mathematics*
*University of Kentucky*

**Yousef Saad**                                                                        *saad@umn.edu*
*Department of Computer Science*
*University of Minnesota Twin Cities*

**Yuanzhe Xi**                                                                    *yuanzhe.xi@emory.edu*
*Department of Mathematics*
*Emory University*

**Reviewed on OpenReview:** *https://openreview.net/forum?id=vo8FOBt6f6*

## Abstract

Stochastic Gradient Descent (SGD) often slows in the late stage of training due to anisotropic curvature and gradient noise. We analyze preconditioned SGD in the geometry induced by a symmetric positive definite matrix $\mathbf{M}$. Our bounds make explicit how both the convergence rate and the stochastic noise floor depend on $\mathbf{M}$. For nonconvex objectives, we establish a basin-stability guarantee in a local $\mathbf{M}$-metric neighborhood around a minimizer set: under local smoothness and a local PL condition, we give an explicit lower bound on the probability that the iterates remain in the basin up to a time horizon. This perspective is particularly relevant in Scientific Machine Learning (SciML), where reaching small training losses under stochastic updates is closely tied to physical fidelity, numerical stability, and constraint satisfaction. Our framework covers both diagonal/adaptive and curvature-aware preconditioners and yields a practical criterion: choose $\mathbf{M}$ to improve local conditioning while attenuating noise in the $\mathbf{M}^{-1}$-norm. Experiments on a quadratic diagnostic and three SciML benchmarks support the predicted rate–floor behavior.

# 1 Introduction

Stochastic Gradient Descent (SGD) has long been the workhorse of large-scale machine learning. Since its early application to multilayer perceptrons in the 1960s (Amari, 1967), its simplicity, scalability, and low per-iteration cost have made it a popular optimizer for deep learning models (Bottou et al., 2018). Classical convergence theory for SGD under noisy gradients typically guarantees a sublinear rate of $\mathcal{O}(1/k)$ under convexity and smoothness assumptions (Robbins & Monro, 1951; Blum, 1954). The theory for SGD convergence under various combinations of conditions is well studied and documented in Garrigos & Gower (2024); Khaled & Richtárik (2023), and Bach (2024).

Recent theoretical developments have established *linear convergence* for SGD under stronger conditions, such as strong convexity, smoothness, and bounded noise (Bottou et al., 2018). When the loss $F$ is $c$-strongly convex, has $L$-Lipschitz gradients, and the learning rate $\alpha$ satisfies $\alpha \leq \mu/(LK_G)$, the iterates $\mathbf{w}_k$ satisfy

$$\mathbb{E}[F(\mathbf{w}_k) - F_*] \leq (1 - \alpha c\mu)^{k-1}\left(F(\mathbf{w}_1) - F_* - \frac{\alpha LK}{2c\mu}\right) + \frac{\alpha LK}{2c\mu}, \tag{1}$$

where $\mu$, $K$, and $K_G$ are constants associated with the stochastic gradients (defined in Assumptions 9–11), and let $\mathbf{w}^*$ denote the unique minimizer and $F_* := F(\mathbf{w}^*)$ the optimal value. Eq. (1) highlights two late-stage drivers: a linear contraction factor $1 - \alpha c\mu$ and a stochastic error floor

$$\frac{\alpha LK}{2c\mu} = \frac{\alpha}{2\mu}\kappa K,$$

where $\kappa := \frac{L}{c}$ is the (Euclidean) condition number associated with curvature. For any admissible $\alpha$, the floor scales with $\kappa$ and $K$, while the contraction depends on the product $\alpha c\mu$.

Many successful optimizers can be viewed as *preconditioned variants of SGD*. Adaptive methods such as Adagrad (Duchi et al., 2011), Adam (Kingma & Ba, 2017), and RMSProp (Hinton, 2014), structured second-order approaches including Shampoo (Gupta et al., 2018), natural gradient descent (Amari, 1998), K-FAC (Martens & Grosse, 2015; Ishikawa & Karakida, 2024), and Sophia (Liu et al., 2024), as well as quasi-Newton methods like L-BFGS (Liu & Nocedal, 1989; Chen et al., 2014), all apply a linear transformation to the gradient that reshapes both curvature and gradient noise. From this perspective, their empirical effectiveness indicates that late-stage optimization is influenced not only by the choice of learning rates, but also by how the preconditioning alters local conditioning and the geometry of stochastic noise. Despite their widespread use, however, there is still no unified theoretical framework that identifies which properties of a preconditioner determine the late-stage convergence rate and the attainable noise floor.

Motivated by this perspective, we study the preconditioned SGD update in the following form

$$\mathbf{w}_{k+1} = \mathbf{w}_k - \alpha_k \mathbf{M}^{-1} g(\mathbf{w}_k, \boldsymbol{\xi}_k), \tag{2}$$

where $\mathbf{M} \succ 0$ is a symmetric positive definite (SPD) matrix that defines the geometry in which both curvature and noise are measured, $g(\mathbf{w}_k, \boldsymbol{\xi}_k) = \nabla_w F_k(\mathbf{w})$ is the stochastic gradient, $\alpha_k$ is the learning rate, $\boldsymbol{\xi}_k$ is an i.i.d. sample drawn at iteration $k$. The standard (vanilla) SGD update is recovered when $\mathbf{M} = \mathbf{I}$. Our goal is not to propose a new optimizer, but to provide a principled framework to analyze and compare preconditioners in the late stage of training.

**Main contributions** We investigate how preconditioning influences the late-stage behavior of SGD within a well-behaved basin of the loss surface. By analyzing preconditioned SGD in the $\mathbf{M}$-induced geometry, we show how rescaling the gradient affects both the convergence rate and the attainable noise floor, and we derive criteria that clarify which properties of a preconditioner matter in the late stage of training.

1. **Preconditioned SGD in the strongly convex baseline.** We extend the classical "linear rate + noise floor" theory for SGD to updates preconditioned by a fixed SPD matrix $\mathbf{M}$. The resulting bounds show that late-stage behavior is controlled by (i) an effective conditioning in the $\mathbf{M}$-geometry and (ii) the preconditioned gradient-noise level; the attainable error floor scales with their *product*. Since admissible constant stepsizes are limited by $\mathbf{M}$-smoothness, improved conditioning allows larger stepsizes and hence faster contraction. With diminishing stepsizes, we obtain an $\mathcal{O}(1/k)$ rate.

2. **Local nonconvex regime with basin stability.** Under a local $\mathbf{M}$–PL condition and local smoothness, we establish late-stage convergence guarantees inside a well-behaved basin around a minimizer set, again with an explicit rate–floor structure. In addition, we provide a basin-stability bound that lower-bounds the probability of remaining in the basin up to a horizon.

3. **Design criteria and empirical evidence.** Our theory yields a simple design principle: choose $\mathbf{M}$ to improve local conditioning while attenuating noise in the $\mathbf{M}^{-1}$-norm; the attainable late-stage floor tracks their product. We validate this mechanism on (i) a quadratic diagnostic where the relevant constants can be computed in closed form, and (ii) three SciML benchmarks where late-stage behavior is strongly tied to final accuracy.

While late-stage convergence is broadly relevant, it is especially important in SciML. Here, training losses encode physically meaningful quantities (e.g., PDE residuals, boundary conditions, stability). Unlike standard ML tasks where moderate error may still be acceptable, small reductions in the final loss can determine whether solutions conserve invariants, remain stable over long horizons, or meet scientific accuracy requirements. In this setting, the optimizer's asymptotic behavior—and particularly the final noise floor—directly governs physical fidelity (Rathore et al., 2024).

## 2    Related work

Recent work has advanced the theoretical understanding of preconditioned and adaptive variants of SGD under various structural and noise assumptions. Koren et al. (2022) showed that preconditioned SGD achieves a rate of $\mathcal{O}(1/\sqrt{k})$ for general stochastic convex optimization, though convergence can stagnate in the presence of persistent gradient noise. Faw et al. (2022) further established that adaptive SGD attains an order-optimal $\tilde{\mathcal{O}}(1/\sqrt{k})$ rate for nonconvex smooth objectives under affine variance conditions, without requiring bounded gradients or finely tuned learning rates. More recently, Attia & Koren (2023) derived high-probability guarantees of $\tilde{\mathcal{O}}(1/k + \sigma_0/\sqrt{k})$ for adaptive methods in both convex and nonconvex settings, relaxing the need for strong smoothness or prior parameter knowledge.

These results primarily address *global* convergence behavior across general problem classes. In contrast, our analysis focuses on the *asymptotic regime*—the late stage of training where iterates lie within a well-behaved basin around a local minimizer and optimization progress is limited by curvature anisotropy and gradient noise. In this regime, we show that both the convergence rate and the noise floor of the preconditioned SGD are determined by curvature and variance quantities measured in the preconditioned geometry. This local, geometry-aware viewpoint clarifies why curvature-informed preconditioners and adaptive algorithms yield faster and more stable late-stage convergence.

Other techniques such as batch normalization (Lange et al., 2022) and weight decay (Loshchilov & Hutter, 2017; Barrett & Dherin, 2020) can also be interpreted as implicit forms of preconditioning, though they operate through different regularization mechanisms. For comprehensive surveys of explicit preconditioned SGD and related adaptive methods, we refer the reader to Ye (2024). Beyond convergence rates, preconditioning has also been studied as an implicit regularization that may affect generalization (Amari et al., 2021). Our paper, however, focuses on optimization and training loss rather than test error or generalization. This emphasis is deliberate in many SciML problems, where the training objective often directly measures physically meaningful quantities such as PDE residuals and boundary-condition violations, so driving the training loss low is itself important. At the same time, preconditioning changes the optimization trajectory and therefore the algorithm's implicit bias, so it may also affect generalization. Understanding how the metric-dependent quantities in our analysis interact with out-of-sample accuracy is an important direction for future work.

## 3    Preconditioned SGD convergence analysis

We first analyze the globally strongly convex case as a *baseline* to make the role of the preconditioned geometry explicit. Although this setting is rarely realized in deep learning, it reveals the essential mechanism through which preconditioning affects convergence. The analysis shows how curvature and noise floor transform

under a change of metric, providing a principled way to compare different choices of $\mathbf{M}$. This also lays the groundwork for the local nonconvex analysis in Section 3.2, where $\mathbf{M}$ influences both basin size and stability.

### 3.1 Convergence in the globally strongly convex setting

We establish convergence guarantees for preconditioned SGD when the objective is globally strongly convex. This simplified setting allows for a transparent analysis of how a preconditioner reshapes both the effective curvature and the gradient noise. While the derivations parallel the Euclidean case, expressing them in the $\mathbf{M}$-induced geometry makes the dependence on the preconditioner explicit and lays the groundwork for the more general nonconvex results to follow.

**Curvature assumptions.** Preconditioning redefines smoothness and strong convexity through effective constants $(\hat{L}, \hat{c})$ measured in the $\mathbf{M}$–induced norm.

**Assumption 1** ($\mathbf{M}$-strong convexity). *$F : \mathbb{R}^d \to \mathbb{R}$ is $\mathbf{M}$-strongly convex: there exists $\hat{c} > 0$ such that*

$$F(\overline{\mathbf{w}}) \geq F(\mathbf{w}) + \nabla F(\mathbf{w})^\top (\overline{\mathbf{w}} - \mathbf{w}) + \tfrac{1}{2}\,\hat{c}\,\|\overline{\mathbf{w}} - \mathbf{w}\|_\mathbf{M}^2, \quad \forall\, \overline{\mathbf{w}}, \mathbf{w} \in \mathbb{R}^d.$$

**Assumption 2** ($\mathbf{M}$-Lipschitz gradient). *$\nabla F$ is $\mathbf{M}$-Lipschitz with constant $\hat{L} > 0$:*

$$\|\nabla F(\overline{\mathbf{w}}) - \nabla F(\mathbf{w})\|_{\mathbf{M}^{-1}} \leq \hat{L}\,\|\overline{\mathbf{w}} - \mathbf{w}\|_\mathbf{M}, \quad \forall\, \overline{\mathbf{w}}, \mathbf{w} \in \mathbb{R}^d.$$

These conditions are direct analogues of the Euclidean definitions. Writing $\mathbf{M}^{-1} = \mathbf{P}\mathbf{P}^\top$ gives the spectral characterization:

**Lemma 3.1.** *Let $F$ be twice differentiable and $\mathbf{M}^{-1} = \mathbf{P}\mathbf{P}^\top$. Then: (i) $\nabla F$ is $\mathbf{M}$-Lipschitz with constant $\hat{L} \iff$ all eigenvalues of $\mathbf{P}^\top \nabla^2 F(\mathbf{w})\mathbf{P}$ are $\leq \hat{L}$; (ii) $F$ is $\mathbf{M}$-strongly convex with constant $\hat{c} \iff$ all eigenvalues of $\mathbf{P}^\top \nabla^2 F(\mathbf{w})\mathbf{P}$ are $\geq \hat{c}$.*

Hence, preconditioning improves the effective condition number whenever $\hat{L}/\hat{c} < L/c$.

**Noise assumptions.** We measure the first and second moments of the stochastic gradient in the $\mathbf{M}^{-1}$–norm. Specifically, holding $\mathbf{w}_k$ fixed, we define the variance with respect to the sampling of $\boldsymbol{\xi}_k$ by

$$\mathbb{V}_{\boldsymbol{\xi}_k}\big[g(\mathbf{w}_k, \boldsymbol{\xi}_k), \|\cdot\|_{\mathbf{M}^{-1}}\big] := \mathbb{E}_{\boldsymbol{\xi}_k}\big[\|g(\mathbf{w}_k, \boldsymbol{\xi}_k)\|_{\mathbf{M}^{-1}}^2\big] - \big\|\mathbb{E}_{\boldsymbol{\xi}_k}[g(\mathbf{w}_k, \boldsymbol{\xi}_k)]\big\|_{\mathbf{M}^{-1}}^2. \qquad (3)$$

**Assumption 3** (Moment bounds in $\mathbf{M}^{-1}$). *For the iterates of (2), there exist constants $\mu_G \geq \mu > 0$, $K \geq 0$, and $K_V \geq 0$ such that, for all $k$,*

$$\langle \nabla F(\mathbf{w}_k),\, \mathbb{E}_{\boldsymbol{\xi}_k}[g(\mathbf{w}_k, \boldsymbol{\xi}_k)]\rangle_{\mathbf{M}^{-1}} \geq \mu\,\|\nabla F(\mathbf{w}_k)\|_{\mathbf{M}^{-1}}^2, \qquad (4)$$

$$\|\mathbb{E}_{\boldsymbol{\xi}_k}[g(\mathbf{w}_k, \boldsymbol{\xi}_k)]\|_{\mathbf{M}^{-1}} \leq \mu_G\,\|\nabla F(\mathbf{w}_k)\|_{\mathbf{M}^{-1}}, \qquad (5)$$

$$\mathbb{V}_{\boldsymbol{\xi}_k}\big[g(\mathbf{w}_k, \boldsymbol{\xi}_k), \|\cdot\|_{\mathbf{M}^{-1}}\big] \leq K + K_V\,\|\nabla F(\mathbf{w}_k)\|_{\mathbf{M}^{-1}}^2. \qquad (6)$$

We call $K$ the *preconditioned noise level* because the variance in the $\mathbf{M}^{-1}$–norm satisfies

$$\mathbb{V}_{\boldsymbol{\xi}}\big[g(\mathbf{w}, \boldsymbol{\xi}), \|\cdot\|_{\mathbf{M}^{-1}}\big] = \operatorname{tr}(\mathbf{M}^{-1}\boldsymbol{\Sigma}(\mathbf{w})),$$

where $\boldsymbol{\Sigma}(\mathbf{w}) := \operatorname{Cov}(g(\mathbf{w}, \boldsymbol{\xi}) \mid \mathbf{w})$. In the stationary case $\boldsymbol{\Sigma}(\mathbf{w}) \equiv \boldsymbol{\Sigma}$, we have the fixed $\operatorname{tr}(\mathbf{M}^{-1}\boldsymbol{\Sigma})$. More generally, on a region containing the iterates it is natural to choose $K \geq \sup_\mathbf{w} \operatorname{tr}(\mathbf{M}^{-1}\boldsymbol{\Sigma}(\mathbf{w}))$, so $K$ is a uniform baseline for the preconditioned noise.

Under these assumptions we obtain the usual linear and sublinear rates, but with constants that depend explicitly on the preconditioned geometry.

**Theorem 3.2.** *Under Assumptions 1–3 (with $F_{\min} = F_*$), suppose (2) uses a fixed learning rate $\alpha_k = \overline{\alpha}$ with*

$$0 < \overline{\alpha} \leq \frac{\mu}{\hat{L}\,K_G} \qquad \text{where} \ \ K_G = K_V + \mu_G^2 \geq \mu^2 > 0.$$

*Then, for all $k \in \mathbb{N}$,*

$$\mathbb{E}[F(\mathbf{w}_k) - F_*] \leq \frac{\overline{\alpha} \hat{L} K}{2\hat{c}\mu} + (1 - \overline{\alpha}\hat{c}\mu)^{k-1}\left(F(\mathbf{w}_1) - F_* - \frac{\overline{\alpha}\hat{L}K}{2\hat{c}\mu}\right) \xrightarrow{k\to\infty} \frac{\overline{\alpha}\hat{L}K}{2\hat{c}\mu}. \tag{7}$$

Theorem 3.2 shows that, with a fixed learning rate $\overline{\alpha}$, preconditioned SGD contracts linearly with factor $1 - \overline{\alpha}\hat{c}\mu$ and converges to an asymptotic floor

$$\frac{\overline{\alpha}\hat{L}K}{2\hat{c}\mu} = \frac{\overline{\alpha}}{2\mu}\left(\frac{\hat{L}}{\hat{c}}\right)K.$$

Thus, the floor factorizes into an *effective condition number* $\hat{L}/\hat{c}$ and a *preconditioned noise level $K$*. In the late stage of training, we have $F(\mathbf{w}_k) - F_* = \mathcal{O}(\overline{\alpha}K)$ and $\|\nabla F(\mathbf{w}_k)\|_{\mathbf{M}^{-1}}^2 = \mathcal{O}(\overline{\alpha}K)$. Substituting into the variance bound (6) gives

$$\mathbb{V}_{\boldsymbol{\xi}_k}\big[g(\mathbf{w}_k, \boldsymbol{\xi}_k), \|\cdot\|_{\mathbf{M}^{-1}}\big] \leq K + \mathcal{O}(\overline{\alpha}K),$$

so for small $\overline{\alpha}$ the variance is dominated by the baseline $K$ term.

Moreover, since $\mathbb{V}_{\boldsymbol{\xi}}[g(\mathbf{w}, \boldsymbol{\xi}), \|\cdot\|_{\mathbf{M}^{-1}}] = \text{tr}(\mathbf{M}^{-1}\boldsymbol{\Sigma}(\mathbf{w}))$, we may view $K$ as an upper baseline for the preconditioned noise $\text{tr}(\mathbf{M}^{-1}\boldsymbol{\Sigma}(\mathbf{w}))$ along the late–stage trajectory. Preconditioning reduces this baseline through its effect on $\text{tr}(\mathbf{M}^{-1}\boldsymbol{\Sigma}(\mathbf{w}))$; choosing $\mathbf{M}$ to attenuate high–variance directions lowers this trace and thus lowers the effective noise floor.

**Theorem 3.3.** *Under Assumptions 1–3 (with $F_{\min} = F_*$), suppose (2) uses $\alpha_k = \beta/(\gamma + k)$ with $\beta > \frac{1}{\hat{c}\mu}$ and $\gamma > 0$ chosen so that $\alpha_1 \leq \mu/(\hat{L}K_G)$. Then, for all $k \in \mathbb{N}$,*

$$\mathbb{E}[F(\mathbf{w}_k) - F_*] \leq \frac{\nu}{\gamma + k}, \qquad \nu := \max\left\{\frac{\beta^2 \hat{L}K}{2(\beta\hat{c}\mu - 1)}, (\gamma + 1)\big(F(\mathbf{w}_1) - F_*\big)\right\}. \tag{8}$$

With diminishing learning rates, the noise floor vanishes and Theorem 3.3 shows that preconditioned SGD attains the optimal $\mathcal{O}(1/k)$ rate. Preconditioning no longer changes the rate itself—it always decays like $1/k$—but it directly influences the leading constant $\nu$ which has the same structure as the fixed-learning-rate floor: an effective condition number $\hat{L}/\hat{c}$ multiplied by the preconditioned noise level $K$. Thus even when the noise floor disappears, late–stage performance is still governed by the same metric–dependent quantities $(\hat{L}, \hat{c}, K)$. Consequently, effective preconditioners must again balance curvature alignment (to reduce $\hat{L}/\hat{c}$) with noise attenuation (to reduce $K$), improving both the asymptotic constants in the $\mathcal{O}(1/k)$ regime.

## 3.2 Local convergence in the nonconvex setting

The empirical loss $F(\mathbf{w})$ over network parameters is typically *nonconvex*, and its local geometry near minimizers is rarely strictly convex. Empirical studies show that trained models often converge to regions that are flat in many directions and exhibit highly degenerate curvature—manifested as a cluster of very small or near-zero eigenvalues in the Hessian—arising from overparameterization, symmetries, and parameter non-identifiability (Sagun et al., 2018; Ghorbani et al., 2019). Despite this degeneracy, the optimization dynamics remain structured: iterates contract along directions with significant curvature while the loss changes little along flat directions. To describe this late-stage regime without assuming strong convexity, we impose a *local Polyak–Łojasiewicz (PL)* condition (Chan, 1979; Karimi et al., 2016) in the $\mathbf{M}$–geometry, which enforces gradient domination only in informative directions and tolerates flat or weakly curved subspaces. This flat-tolerant formulation provides a natural framework to study how preconditioning reshapes local curvature and noise, governing contraction rates, asymptotic error floors, and stability during the final phase of optimization.

**Additional local assumptions.** Fix an SPD matrix $\mathbf{M}$ and an open neighborhood $\mathcal{U} \subset \mathbb{R}^d$. Assume the local minimizer set

$$\mathcal{S} := \arg\min_{\mathbf{w} \in \mathcal{U}} F(\mathbf{w}) \neq \varnothing, \qquad F_* := \min_{\mathbf{w} \in \mathcal{U}} F(\mathbf{w}) = F(\mathbf{s}) \text{ for any } \mathbf{s} \in \mathcal{S}.$$

Write $\|x\|_{\mathbf{M}} := (x^\top \mathbf{M} x)^{1/2}$ and $\mathrm{dist}_{\mathbf{M}}(\mathbf{w}, \mathcal{S}) := \inf_{\mathbf{s} \in \mathcal{S}} \|\mathbf{w} - \mathbf{s}\|_{\mathbf{M}}$. For radii $0 < r < r_+$, define the $\mathbf{M}$–metric neighborhoods

$$\mathcal{N}_r := \{\mathbf{w} : \mathrm{dist}_{\mathbf{M}}(\mathbf{w}, \mathcal{S}) \le r\}, \qquad \mathcal{N}_{r_+} := \{\mathbf{w} : \mathrm{dist}_{\mathbf{M}}(\mathbf{w}, \mathcal{S}) \le r_+\} \subseteq \mathcal{U}.$$

We assume the following conditions hold on $\mathcal{N}_r$ (for the iterates) and on $\mathcal{N}_{r_+}$ (for the exit bound).

**Assumption 4** (Local $\mathbf{M}$–PL on $\mathcal{N}_r$)**.** *There exists $\hat{\mu}_{\mathrm{PL}} > 0$ such that, for all $\mathbf{w} \in \mathcal{N}_r$,*

$$2\hat{\mu}_{\mathrm{PL}}\big(F(\mathbf{w}) - F_*\big) \le \|\nabla F(\mathbf{w})\|_{\mathbf{M}^{-1}}^2.$$

**Assumption 5** (Local $\mathbf{M}$–Lipschitz gradient on a convex neighborhood of $\mathcal{N}_{r_+}$)**.** *There exists an open convex set $\mathcal{V}$ with $\mathcal{N}_{r_+} \subset \mathcal{V} \subseteq \mathcal{U}$ and a constant $\hat{L} > 0$ such that, for all $\overline{\mathbf{w}}, \mathbf{w} \in \mathcal{V}$,*

$$\|\nabla F(\overline{\mathbf{w}}) - \nabla F(\mathbf{w})\|_{\mathbf{M}^{-1}} \le \hat{L}\,\|\overline{\mathbf{w}} - \mathbf{w}\|_{\mathbf{M}}.$$

**Assumption 6** (Local stochastic gradient conditions on $\mathcal{N}_r$)**.** *Let $(\mathcal{F}_k)$ denote the natural filtration and set $g_k := g(\mathbf{w}_k, \boldsymbol{\xi}_k)$. There exist constants $\mu \in (0, 1]$, $K_G \ge 0$, and $K \ge 0$ such that, for every $k$ with $\mathbf{w}_k \in \mathcal{N}_r$,*

$$\big\langle \nabla F(\mathbf{w}_k), \mathbb{E}[g_k \mid \mathcal{F}_k] \big\rangle_{\mathbf{M}^{-1}} \ge \mu\,\|\nabla F(\mathbf{w}_k)\|_{\mathbf{M}^{-1}}^2, \qquad \mathbb{E}\big[\|g_k\|_{\mathbf{M}^{-1}}^2 \mid \mathcal{F}_k\big] \le K_G\,\|\nabla F(\mathbf{w}_k)\|_{\mathbf{M}^{-1}}^2 + K.$$

**Assumption 7** (Local quadratic growth (QG) on $\mathcal{N}_{r_+}$)**.** *There exists $\alpha_{\mathrm{QG}} > 0$ such that, for all $\mathbf{w} \in \mathcal{N}_{r_+}$,*

$$F(\mathbf{w}) - F_* \ge \tfrac{\alpha_{\mathrm{QG}}}{2}\,\mathrm{dist}_{\mathbf{M}}(\mathbf{w}, \mathcal{S})^2.$$

**Assumption 8** (Controlled one-step overshoot on $\mathcal{N}_r$)**.** *Fix radii $0 < r < r_+$ and a horizon $T \ge 1$, and set $\Delta := r_+ - r$. There exist deterministic numbers $(\delta_k)_{k=1}^{T-1}$ with $\delta_k \in [0, 1)$ such that for every $k \le T - 1$,*

$$\mathbf{1}_{\{\mathbf{w}_k \in \mathcal{N}_r\}}\,\alpha_k^2\,\mathbb{E}\big[\|g_k\|_{\mathbf{M}^{-1}}^2 \mid \mathcal{F}_k\big] \le \delta_k\,\Delta^2 \qquad a.s.$$

Lemma 3.4 gives the one-step containment probability implied by Assumption 8.

**Lemma 3.4** (Containment probability implied by Assumption 8)**.** *Under Assumption 8, for every $k \le T - 1$,*

$$\mathbf{w}_k \in \mathcal{N}_r \quad \Longrightarrow \quad \mathbb{P}\big(\mathbf{w}_{k+1} \in \mathcal{N}_{r_+} \mid \mathcal{F}_k\big) \ge 1 - \delta_k.$$

These local assumptions are the basin–restricted analogue of the global conditions in Section 3.1. The local $\mathbf{M}$–PL condition replaces global strong convexity by a *gradient–domination* inequality in the $\mathbf{M}$–metric: it enforces curvature only in directions that drive descent while permitting flat or weakly curved directions. The local $\mathbf{M}$–Lipschitz gradient assumption on a convex neighborhood $\mathcal{V} \supset \mathcal{N}_{r_+}$ provides a quadratic upper model along any update segment that stays in $\mathcal{V}$:

$$F(\overline{\mathbf{w}}) \le F(\mathbf{w}) + \nabla F(\mathbf{w})^\top (\overline{\mathbf{w}} - \mathbf{w}) + \tfrac{\hat{L}}{2}\,\|\overline{\mathbf{w}} - \mathbf{w}\|_{\mathbf{M}}^2.$$

In our finite-horizon analysis, this condition is invoked only on trajectories for which the iterates (and hence the corresponding update segments, by convexity) remain inside $\mathcal{V}$ up to time $T$.

The local stochastic gradient condition (Assumption 6) mirrors the global moment bounds in Assumption 3, but is only required to hold when $\mathbf{w}_k \in \mathcal{N}_r$. It imposes a first-moment alignment condition and a *second-moment* bound in the $\mathbf{M}^{-1}$–norm, which is the natural scale for preconditioned updates. The local QG condition ensures that the objective grows at least quadratically with $\mathrm{dist}_{\mathbf{M}}(\mathbf{w}, \mathcal{S})$ near the basin boundary—a property that holds, for example, when curvature is positive in normal directions—and it supplies the barrier needed in the optional-stopping/exit-time argument.

Assumption 8 controls rare one-step overshoots from the inner basin $\mathcal{N}_r$ to outside the enlarged neighborhood $\mathcal{N}_{r_+}$. When $\mathbf{w}_k \in \mathcal{N}_r$, the preconditioned update moves a distance

$$\|\mathbf{w}_{k+1} - \mathbf{w}_k\|_{\mathbf{M}} = \alpha_k\,\|g_k\|_{\mathbf{M}^{-1}}.$$

Since $\text{dist}_{\mathbf{M}}(\mathbf{w}_k, \mathcal{S}) \leq r$ on $\mathcal{N}_r$, the triangle inequality implies that $\mathbf{w}_{k+1} \notin \mathcal{N}_{r_+}$ can occur only if $\alpha_k \|g_k\|_{\mathbf{M}^{-1}} > \Delta$ with $\Delta := r_+ - r$. Assumption 8 bounds the conditional second moment of $\|g_k\|_{\mathbf{M}^{-1}}$ relative to $\Delta$; therefore, by Markov's inequality,

$$\mathbb{P}(\mathbf{w}_{k+1} \notin \mathcal{N}_{r_+} \mid \mathcal{F}_k) \leq \delta_k \qquad \text{whenever } \mathbf{w}_k \in \mathcal{N}_r.$$

Together, these assumptions describe a local regime that accommodates moderate nonconvexity and flatness while still providing sufficient structure for quantitative finite-horizon convergence and stability guarantees under stochastic gradients.

**Theorem 3.5** (Convergence in a local basin up to a finite horizon). *Fix radii $0 < r < r_+$ and a horizon $T \geq 1$, and let*

$$\tau := \inf\{k \geq 1 : \mathbf{w}_k \notin \mathcal{N}_r\}, \qquad \Omega_T := \{\tau > T\}.$$

*Assume: (i) Assumptions 4 and 6 hold on $\mathcal{N}_r$; (ii) Assumption 5 holds on a convex set $\mathcal{V}$ with $\mathcal{N}_{r_+} \subset \mathcal{V} \subseteq \mathcal{U}$; (iii) Assumption 7 holds on $\mathcal{N}_{r_+}$; (iv) Assumption 8 holds with horizon $T$ and failure probabilities $(\delta_k)_{k=1}^{T-1}$; and (v) the conditional-moment version of Assumption 6 holds on $\Omega_T$ (i.e., the first/second-moment bounds are valid when conditioning on $(\mathcal{F}_k, \Omega_T)$ for $k \leq T-1$).*

*Suppose $\mathbf{w}_1 \in \mathcal{N}_r$ and use a constant stepsize $\alpha_k = \overline{\alpha}$ such that*

$$0 < \overline{\alpha} \leq \frac{\mu}{\hat{L} K_G} \quad (\text{if } K_G > 0), \qquad \text{and} \qquad 0 < \overline{\alpha} < \frac{1}{\mu \hat{\mu}_{\text{PL}}}.$$

*Define*

$$\rho := \overline{\alpha} \, \hat{\mu}_{\text{PL}} \, \mu \in (0, 1), \qquad C := \frac{\overline{\alpha} \, \hat{L} \, K}{2 \, \hat{\mu}_{\text{PL}} \, \mu}, \qquad B := \frac{\alpha_{\text{QG}}}{2} \, r^2.$$

*For all $1 \leq k \leq T$,*

$$\mathbb{E}[F(\mathbf{w}_k) - F_* \mid \tau > T] \leq C + (1 - \rho)^{k-1} (F(\mathbf{w}_1) - F_* - C).$$

*The probability of remaining in $\mathcal{N}_r$ up to time $T$ satisfies*

$$\mathbb{P}(\tau > T) \geq \left[ 1 - \frac{F(\mathbf{w}_1) - F_* + \frac{\hat{\ell}}{2} \overline{\alpha}^2 K (T-1)}{B} - \sum_{k=1}^{T-1} \delta_k \right]_+,$$

*where $[x]_+ := \max\{0, x\}$.*

**Theorem 3.6** (Diminishing learning rate in a local basin up to a finite horizon). *Fix radii $0 < r < r_+$ and a horizon $T \geq 1$, and let*

$$\tau := \inf\{k \geq 1 : \mathbf{w}_k \notin \mathcal{N}_r\}, \qquad \Omega_T := \{\tau > T\}.$$

*Assume: (i) Assumptions 4 and 6 hold on $\mathcal{N}_r$; (ii) Assumption 5 holds on a convex set $\mathcal{V}$ with $\mathcal{N}_{r_+} \subset \mathcal{V} \subseteq \mathcal{U}$; (iii) Assumption 7 holds on $\mathcal{N}_{r_+}$; (iv) Assumption 8 holds with horizon $T$ and failure probabilities $(\delta_k)_{k=1}^{T-1}$; and (v) the conditional-moment version of Assumption 6 holds on $\Omega_T$.*

*Suppose $\mathbf{w}_1 \in \mathcal{N}_r$ and use harmonic stepsizes*

$$\alpha_k = \frac{\beta}{\gamma + k}, \qquad \gamma > 0,$$

*with*

$$0 < \alpha_1 = \frac{\beta}{\gamma + 1} \leq \frac{\mu}{\hat{L} K_G} \quad (\text{if } K_G > 0), \qquad \text{and} \qquad \beta > \frac{1}{\mu \hat{\mu}_{\text{PL}}} \quad (\text{equivalently } a := \beta \mu \hat{\mu}_{\text{PL}} > 1).$$

*Define*

$$m := \mu \hat{\mu}_{\text{PL}}, \qquad c := \frac{\hat{L} K}{2}, \qquad B := \frac{\alpha_{\text{QG}}}{2} \, r^2, \qquad \nu := \max\left\{ \frac{c \beta^2}{\beta m - 1}, \; (\gamma + 1) \left[ F(\mathbf{w}_1) - F_* \right] \right\}.$$

*For all $1 \leq k \leq T$,*

$$\mathbb{E}[F(\mathbf{w}_k) - F_* \mid \tau > T] \leq \frac{\nu}{\gamma + k}.$$

*The probability of remaining in $\mathcal{N}_r$ up to time $T$ satisfies*

$$\mathbb{P}(\tau > T) \geq \left[1 - \frac{F(\mathbf{w}_1) - F_* + c \sum_{k=1}^{T-1} \alpha_k^2}{B} - \sum_{k=1}^{T-1} \delta_k\right]_+,$$

*where $[x]_+ := \max\{0, x\}$.*

Theorem 3.5 (fixed stepsize) and Theorem 3.6 (harmonic stepsizes) characterize late-stage optimization *after* the iterates have entered a well-behaved local basin $\mathcal{N}_r$. Both results are stated on the finite-horizon survival event

$$\Omega_T := \{\tau > T\}, \qquad \tau := \inf\{k \geq 1 : \mathbf{w}_k \notin \mathcal{N}_r\},$$

so that along $\Omega_T$ the local $\mathbf{M}$–smoothness and local $\mathbf{M}$–PL inequalities apply to the entire trajectory up to time $T$ and yield explicit descent recursions. With a constant stepsize $\overline{\alpha}$, Theorem 3.5 gives conditional geometric contraction to the noise floor $C = \frac{\overline{\alpha} \hat{L} K}{2 \hat{\mu}_{\mathrm{PL}} \mu}$, whereas with harmonic stepsizes $\alpha_k = \beta/(\gamma + k)$, Theorem 3.6 yields the conditional $\mathcal{O}(1/k)$ rate. In both cases, the constants are *local* and expressed in the $\mathbf{M}$–geometry. Unlike global strongly convex analyses, no global curvature or global variance control is required; the bounds depend only on the basin actually explored by the iterates.

The basin-stability guarantees are also local, and they make two distinct failure mechanisms explicit. The first is an objective barrier controlled by the local QG constant $\alpha_{\mathrm{QG}}$ and the basin radius $r$ through

$$B := \frac{\alpha_{\mathrm{QG}}}{2} r^2,$$

which quantifies the minimum objective increase needed to reach the boundary $\mathcal{N}_{r_+} \setminus \mathcal{N}_r$. The second is one-step overshoot: Assumption 8 allows rare updates that jump from $\mathcal{N}_r$ to outside the enlarged neighborhood $\mathcal{N}_{r_+}$, with conditional failure probabilities $\delta_k$. Here, $\sum_{k=1}^{T-1} \delta_k$ quantifies the accumulated overshoot risk: if the tails/second moments are large, or if the basin margin $\Delta = r_+ - r$ is small, then $\delta_k$ may be large, and the stability bound becomes conservative.

Because all constants in the local bounds are $\mathbf{M}$–dependent, a well-chosen preconditioner $\mathbf{M}$ can improve late-stage behavior by: (i) enhancing local conditioning (increasing $\hat{\mu}_{\mathrm{PL}}$ and/or decreasing $\hat{L}$, thereby strengthening contraction); (ii) reducing the preconditioned noise level $K$; and (iii) improving stability by reducing the overshoot probabilities $\delta_k$ (e.g., via smaller $\mathbb{E}[\|g_k\|_{\mathbf{M}^{-1}}^2]$ and/or a larger margin $\Delta = r_+ - r$) and, when aligned with normal-space curvature, by increasing the barrier parameter $B = \frac{\alpha_{\mathrm{QG}}}{2} r^2$.

## 3.3 Practical preconditioners for SGD

A wide range of preconditioning strategies are used in modern machine learning. On the first–order side, adaptive methods such as Adam (Kingma & Ba, 2017), AMSGrad (Reddi et al., 2018), PAdam (Chen et al., 2020), and Yogi (Zaheer et al., 2018) implicitly apply *diagonal* preconditioners by rescaling gradients with running estimates of coordinatewise second moments. On the second–order side, *curvature-aware* preconditioners exploit Hessian or Fisher Information Matrix (FIM) structure, including the empirical FIM (Schraudolph, 2002), full or mini-batch Hessians (Fletcher, 2013; Garg et al., 2024), mini-batch quasi-Newton updates (Griffin et al., 2022), and Kronecker-factored FIM (K-FAC) (Martens & Grosse, 2015). Classical schemes such as L-BFGS (Liu & Nocedal, 1989; Chen et al., 2014) can also be viewed as low-rank, history-based preconditioners. A particularly important special case is *natural gradient descent*, obtained by choosing $\mathbf{M}$ as the Fisher information matrix (Amari, 1998). In that case, the update follows the local information geometry of the model rather than the Euclidean geometry of parameter space. For exponential-family and least-squares settings, the Fisher matrix is closely related to, and in many cases coincides with, the generalized Gauss–Newton matrix (Schraudolph, 2002; Martens, 2020). Our present theory treats a fixed SPD metric $\mathbf{M}$, so it does not directly analyze the fully time-varying choice $\mathbf{M}_k = \mathbf{F}(\mathbf{w}_k)$.

Nevertheless, it provides a local lens on natural-gradient-type methods.Appendix B.2 summarizes these approaches and their computational trade-offs.

The convergence analysis in Sections 3.1–3.2 suggests two practical mechanisms through which preconditioners shape late-stage behavior:

- *Local conditioning.* Curvature-aware preconditioners (e.g., Fisher, Gauss–Newton, Hessian, K-FAC) tend to reduce the metric–smoothness constant $\hat{L}$ and can increase the local PL constant $\hat{\mu}_{\mathrm{PL}}$. In our bounds, this improves the effective local condition number $\hat{L}/\hat{\mu}_{\mathrm{PL}}$, permits larger admissible fixed learning rates $\alpha \leq \mu/(\hat{L}K_G)$, and reduces the leading constant under diminishing learning rates.

- *Noise attenuation.* Preconditioners aligned with the gradient-noise structure reduce the preconditioned noise level $K$ in the late-stage regime. Together with improved conditioning (smaller $\hat{L}/\hat{c}$ or $\hat{L}/\hat{\mu}_{\mathrm{PL}}$), this lowers the noise floor, which scales with their *product*. Fisher-based and related methods are especially effective because they explicitly incorporate gradient statistics.

These two mechanisms—improved conditioning and reduced preconditioned noise—match the behavior observed in Section 4. Curvature-matched preconditioners (Fisher, Gauss–Newton, K-FAC, Hessian) typically yield faster late-stage contraction by reducing $\hat{L}$ and, in some cases, increasing $\hat{\mu}_{\mathrm{PL}}$, while their use of gradient second-moment information tends to reduce $K$. Adaptive/diagonal methods likewise lower $K$ by damping high-variance coordinates, though their alignment with curvature is typically weaker. Recent theory further suggests that in anisotropic settings, Kronecker-structured preconditioning can be statistically necessary for efficient feature learning, whereas entry-wise/diagonal scaling offers only partial improvements (Zhang et al., 2025).

# 4 Numerical results

Many machine-learning benchmarks illustrate the benefits of preconditioned SGD (e.g., Schmidt et al. (2021); Schneider et al. (2019)), but our emphasis is on SciML, where driving the loss to very small values is tightly linked to physical fidelity, numerical stability, and constraint satisfaction. We therefore structure the experiments in two parts.

First, we analyze a *diagnostic quadratic model* in which all the quantities in our theory—$\hat{L}$, $\hat{\mu}_{\mathrm{PL}}$, and the preconditioned noise level $K$—admit closed forms. This allows us to directly compute the geometry– and noise–dependent metrics from Sections 3.1–3.2 and verify their influence on rate and floor.

Second, we examine three representative SciML problems: noisy Franke surface regression (Franke, 1979), a Poisson–type PINN, and Green's–function learning for diffusion and convection–diffusion (Zhang et al., 2024; Rathore et al., 2024; Hao et al., 2024; Xu et al., 2025), to see how the theoretical mechanisms are reflected in practical settings.

## 4.1 Diagnostic quadratic model

To isolate the effects predicted by the theory, we consider the quadratic objective

$$F(\mathbf{w}) = \tfrac{1}{2}(\mathbf{w} - \mathbf{w}^*)^\top \mathbf{H}(\mathbf{w} - \mathbf{w}^*) + F_*, \qquad \mathbf{H} \succeq 0,$$

here $\mathbf{H}$ specifies curvature. We test two simple, analytically tractable choices: Euclidean SGD ($\mathbf{M} = \mathbf{I}$) and a low-rank curvature-aware preconditioner $\mathbf{M} = \mathbf{I} + \mathbf{U}_s(\widetilde{\mathbf{\Lambda}}_s - \mathbf{I})\mathbf{U}_s^\top$, where $\mathbf{U}_s$ contains the top (or bottom) $s$ eigenvectors of $\mathbf{H}$ and $\widetilde{\mathbf{\Lambda}}_s$ is a diagonal matrix. This model captures the essential effect of curvature information. We used a fixed learning rate.

Instead of forming a dataset, we synthesize unbiased mini–batch gradients

$$g_k = \nabla F(\mathbf{w}_k) + \zeta_k, \qquad \mathbb{E}[\zeta_k] = 0, \quad \mathrm{Cov}(\zeta_k) = \frac{1}{B}\mathbf{\Sigma}.$$

To match the second-order statistics of least-squares problems near $\mathbf{w}^*$, we set $\boldsymbol{\Sigma} = \sigma^2 \mathbf{H}$, giving $K = \frac{\sigma^2}{B} \operatorname{tr}(\mathbf{M}^{-1}\mathbf{H})$. We choose $d = 100$ and construct $\mathbf{H} = \mathbf{U}\boldsymbol{\Lambda}\mathbf{U}^\top$ with $\boldsymbol{\Lambda}$ log-uniform grid on $[10^{-2}, 10^2]$ and $\mathbf{U}$ Haar-distributed. We set $\mathbf{w}^* = 0$, $F_* = 0$, and initialize $\mathbf{w}_1 \sim \mathcal{N}(0, 10^{-4}\mathbf{I})$, and report averages over 30 runs. To illustrate how individual eigenvalues affect constants $(\hat{L}, \hat{\mu}_{\mathrm{PL}}, K)$, we design three groups of tests targeting different part of the spectrum of $\mathbf{H}$.

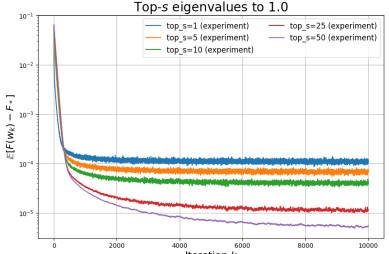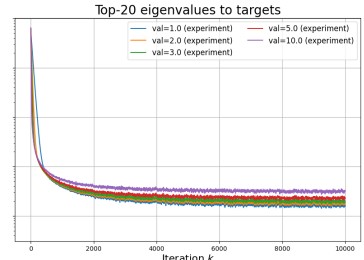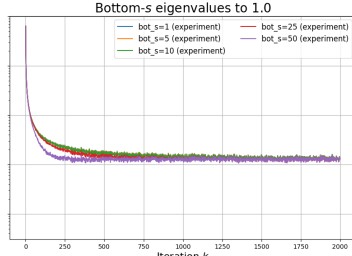

Figure 1: Convergence behavior under different deflation-based preconditioners. Left: deflating the largest $s$ eigenvalues ($s \in \{1, 5, 10, 25, 50\}$). Middle: deflating the top 20 eigenvalues to target values $1.0, 2.0, 3.0, 5.0, 10.0$]. Right: deflating the smallest $s$ eigenvalues ($s \in \{1, 5, 10, 25, 50\}$).

Figure 1 shows how deflating different parts of the spectrum of $\mathbf{H}$ affects the key theoretical constants. Denote the eigenpairs of $\mathbf{H}$ as $(\lambda_i, \mathbf{u}_i)$, and let $\mathbf{U}_s$ contain the selected eigenvectors. We construct a spectral preconditioner of the form $\mathbf{M} = \mathbf{I} + \mathbf{U}_s(\widetilde{\boldsymbol{\Lambda}}_s - \mathbf{I})\mathbf{U}_s^\top$, where $\widetilde{\boldsymbol{\Lambda}}_s = \operatorname{diag}(\tau_1, \ldots, \tau_s)$ assigns a target value $\tau_i$ to the $i$-th chosen eigendirection. Deflating the largest $s$ eigenvalues (left panel)—i.e., setting $\tau_i = \lambda_i$ so that these preconditioned eigenvalues become 1—reduces the smoothness constant $\hat{L}$ and the noise level $K = \frac{\sigma^2}{B}\operatorname{tr}(\mathbf{M}^{-1}\mathbf{H})$ while leaving $\hat{\mu}_{\mathrm{PL}}$ unchanged, yielding a lower noise floor.

To isolate the effect of the noise term, the middle panel fixes $\hat{\mu}_{\mathrm{PL}}$. It deflates the top 20 eigenvalues into a common value $v$ lying between $\lambda_{21}$ and $\lambda_d$ by setting $\tau_i = \lambda_i/v$, so that $\hat{L}$ and $\hat{\mu}_{\mathrm{PL}}$ remain unchanged while $K$ varies. The resulting steady-state losses track this change in $K$, in line with the predicted noise-floor scaling. Deflating the smallest $s$ eigenvalues (right panel)—that is, selecting the bottom eigenvectors and assigning target values $\tau_i$ equal to these smallest eigenvalues so that the preconditioned eigenvalues $\lambda_i/\tau_i$ move to 1—does increase $\hat{\mu}_{\mathrm{PL}}$, but it simultaneously enlarges $K$. The two effects counterbalance each other, yielding only modest overall gains, consistent with the predicted noise-floor behavior.

## 4.2 SciML problems

We then briefly summarize the three SciML tasks used to evaluate late–stage optimization behavior under different preconditioners.

**Noisy Franke surface regression.** The Franke function is a classical multiscale benchmark consisting of several Gaussian peaks with heterogeneous length scales. We sample 256 points uniformly in $[0,1]^2$ and corrupt the values with Gaussian noise $\varepsilon \sim \mathcal{N}(0, 10^{-4})$. The combination of multiscale structure and observational noise yields a loss landscape with varying curvature, making it well suited for evaluating how preconditioning affects convergence in practice. The target surface is

$$f(x,y) = 0.75 e^{-\frac{(9x-2)^2 + (9y-2)^2}{4}} + 0.75 e^{-\frac{(9x+1)^2}{49} - \frac{9y+1}{10}} + 0.5 e^{-\frac{(9x-7)^2 + (9y-3)^2}{4}} - 0.2 e^{-(9x-4)^2 - (9y-7)^2}.$$

**Physics–informed neural networks (PINNs).** We train a PINN to solve the 2D Poisson problem

$$-\Delta u = f(x,y) = 8\pi^2 \sin(2\pi x)\sin(2\pi y) \quad \text{in } (0,1)^2, \qquad u = 0 \text{ on } \partial[0,1]^2,$$

whose exact solution is $u(x,y) = \sin(2\pi x)\sin(2\pi y)$. The training set includes 1,000 interior residual points and 200 boundary points. The weighted loss (PDE residual weight 1.0, boundary weight 100.0) produces a challenging composite landscape known to stress first–order methods (Krishnapriyan et al., 2021). The right panel of Fig. 3 visualizes the source term $f(x,y)$.

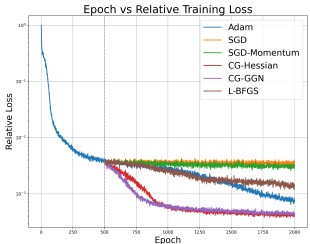 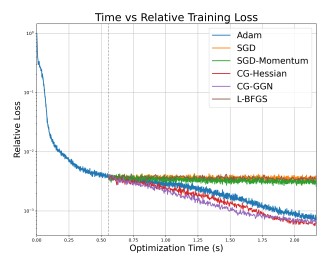 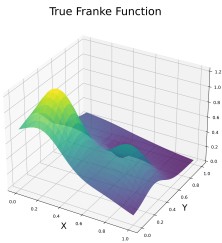

Figure 2: Franke-function regression (mean over 5 runs). Left: relative training loss vs. epochs with the switch to Phase II at epoch 500. Center: relative training loss vs. wall-clock time. Here, relative training loss denotes the training objective normalized by its value at the first plotted epoch, so the initial plotted value is 1.0.
Right: Franke surface.

**Green's–function learning.** We learn Green's functions for the 1D convection–diffusion operator

$$\mathcal{L}u := -\nu u'' + \beta u', \qquad u(0) = u(1) = 0,$$

under two regimes: (i) diffusion-dominated ($\nu = 1.0, \beta = 0$) and (ii) convection-dominated ($\nu = 0.1, \beta = 1.0$). The Green's function satisfies $\mathcal{L}G(x,y) = \delta(x-y)$, where we approximate the delta distribution by a narrow Gaussian with width $\sigma = 0.01$. Training uses: (a) 1,000 uniformly sampled $(x,y)$ pairs for PDE residuals, (b) 500 near-diagonal samples ($|x-y|$ small) to capture the near-singularity, and (c) 200 boundary samples. This produces a highly multiscale and stiffness–dominated optimization problem, ideal for testing curvature-aware preconditioners.

**Baselines and protocol.** Across all SciML tasks, we compare vanilla SGD, momentum, Adam, L–BFGS, and curvature-aware preconditioners (CG–Hessian and CG–GGN/Fisher). Matrix–free CG with a fixed iteration budget is used to apply Hessian or Gauss–Newton/Fisher updates. Following standard SciML practice, we adopt a two-phase schedule: Phase I uses Adam to reach a comparable local basin; Phase II switches to the target optimizer to isolate late-stage behavior. Because our nonconvex theory is local, the basin reached at the end of Phase I can influence the local constants ($\hat{L}, \hat{\mu}_{\mathrm{PL}}, K$) encountered in Phase II and hence may affect which optimizer performs best after the switch. We therefore use the same Adam warm start, switch point, architecture, and seed protocol across all methods to control for basin selection and interpret the Phase II results as comparisons conditional on entering a comparable basin rather than fully basin-agnostic rankings. For all loss-versus-epoch and loss-versus-time plots, we report the *relative training loss*, i.e., the task-specific training objective normalized so that the first plotted epoch has value 1.0. Thus values below 1 indicate reduction relative to the initial training loss. We report loss vs. epochs and wall–clock time, with all architectural and implementation details in Appendix D. All implementations use JAX (Bradbury et al., 2018); code and data are available in the supplemental material.

### 4.3 Noisy data regression

After the Adam warm start (Phase I), Phase II separates the methods (Fig. 2): *Adam, L–BFGS, CG–GGN*, and *CG–Hessian* descend faster than *SGD* and *SGD+Momentum*. The two curvature-aware variants, *CG–Hessian* and *CG–GGN*, track one another closely–showing similar contraction and reaching essentially the same loss floor. The similar performance of *CG-Hessian* and *CG-GGN* suggests that both methods provide comparable normal-space curvature and covariance matrix structure approximation. Adam's diagonal rescaling and *L–BFGS*'s low-rank curvature information also mitigate anisotropy and stabilize noisy directions, which explains their advantage over *SGD*. In wall–clock time, the faster descent of curvature-aware methods compensates for their higher per-step cost.

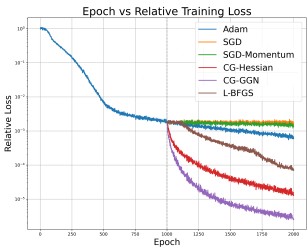 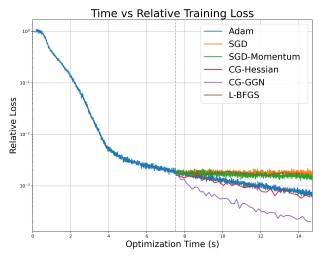 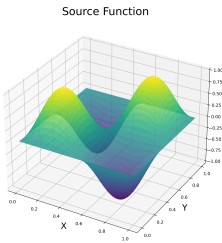

Figure 3: PINN for a Poisson-type PDE (mean over 5 runs). Left: relative training loss vs. epochs with Phase I → Phase II at epoch 1,000. Center: relative training loss vs. wall–clock time.
Right: source term.

## 4.4 Physics–informed neural networks (PINNs)

With the same two-phase protocol, Phase II shows a consistent ranking (Fig. 3). At the bottom, *Adam* and *SGD/SGD+Momentum* lack explicit curvature information and progress slowly. *L-BFGS* achieves intermediate performance: it captures limited curvature through its low-rank approximation and line search, but the memory constraint prevents it from matching the full curvature captured by the two *CG* methods. At the top tier, *CG–GGN* and *CG–Hessian* both achieve better performance as curvature-aware methods, with *CG–GGN* showing a slight advantage.

For PINNs, which minimize weighted least-squares residuals, the Gauss–Newton approximation $\mathbf{J}^\top \mathbf{J}$ is naturally aligned with the gradient covariance structure and thus provides more effective noise attenuation—consistent with our theory, where the preconditioned noise level is governed by $\mathrm{tr}(\mathbf{M}^{-1}\boldsymbol{\Sigma}(\mathbf{w}))$ in the late stage. The Hessian approximation, by contrast, can introduce negative curvature and additional anisotropy. In wall–clock time, *CG–GGN* achieves the best accuracy within a comparable time budget, despite its higher per-step cost.

## 4.5 Green's function learning

After Phase I, Phase II again shows a clear separation of methods (Figs. 4–5). In both the diffusion- and convection–dominated cases, *CG–GGN* continues to drive the loss down, whereas *CG–Hessian*, *L–BFGS*, *Adam*, *SGD*, and *SGD+Momentum* quickly form a tight cluster and improve only marginally. Compared with the earlier PINNs experiment, the Green's–function tasks are more near-singular due to the smoothed-delta forcing, leading to a more challenging, highly anisotropic optimization problem.

Although we did not directly measure the local constants $(\hat{L}, \hat{\mu}_{\mathrm{PL}}, K)$ on this run, the observed advantage of *CG–GGN* is consistent with the structure of PINN objectives. First, for squared-residual losses, the Gauss–Newton/Fisher matrix is positive semidefinite, avoiding the negative-curvature directions introduced by second-derivative terms in the exact Hessian. This makes the preconditioner more stable and better suited to CG. Second, Fisher-type preconditioners are built from gradient second moments and therefore tend to *whiten* gradient noise, reducing the preconditioned noise level $K$. In contrast, a Hessian preconditioner includes second-order terms that are often misaligned with the gradient-noise covariance, and the damping needed to handle indefiniteness diminishes curvature gains while weakening noise attenuation.

These two effects—better alignment with useful curvature and more effective noise whitening—explain why *CG–GGN* reaches lower losses within comparable wall-clock time, despite its higher per-step cost.

The right panels of Figs. 4 and 5 display the learned Green's functions $G(x, y)$ at three representative source locations $y$ together with simple operator and boundary checks for *CG–GGN*. The kernels are localized around the source locations and decay toward the Dirichlet boundaries, and the corresponding operator evaluations produce narrow spikes at $x = y$, in line with the smoothed–delta forcing used in the training loss. This suggests that the lower training losses achieved by *CG–GGN* reflect a reasonable Green's-function approximation rather than a purely numerical artifact.

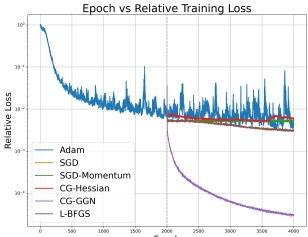 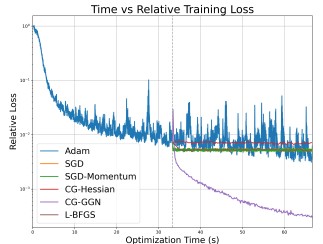 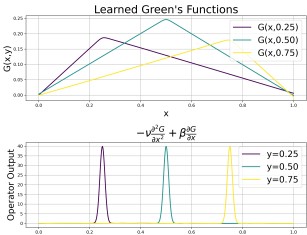

Figure 4: Laplacian Green's function learning (mean over 5 runs). Left: relative training loss vs. epochs with Phase I → Phase II at epoch 2,000. Center: relative training loss vs. wall–clock time.
Right: learned $G(x, y)$ for three source locations and operator checks.

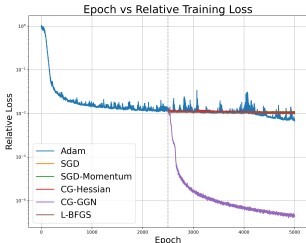 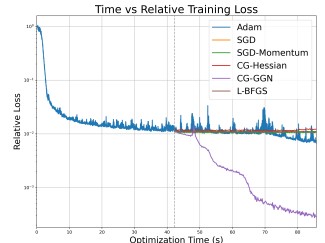 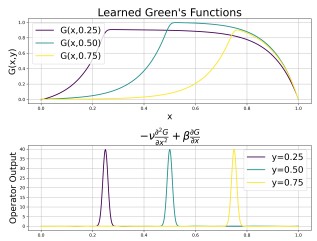

Figure 5: Convection–diffusion Green's function learning (mean over 5 runs). Left: relative training loss vs. epochs with Phase I → Phase II at epoch 2,500. Center: relative training loss vs. wall–clock time.
Right: learned $G(x, y)$ for three source locations and operator checks.

We conclude the numerical experiments by connecting the CG–GGN preconditioner to the theoretical convergence framework developed in this paper. We empirically examine the quantities $L$ and $K$ that govern the convergence of preconditioned SGD for the PINNs problem and two Green's function learning problems. Because a CG-based preconditioner with only a few iterations typically does not significantly alter the cluster of near-zero eigenvalues, we treat the $\mathbf{M}$–PL constant as unchanged and attribute the quality of the preconditioner primarily to its effect on $L$ and $K$. For these three problems, we fix the random seed to 42 and analyze the network parameters at epoch 250 in Phase II. After preconditioning, the $L$ value reduced by factors of 78x, 3710x, and 1923x, respectively. We additionally quantify the impact of preconditioning on the noise level $K$. Using the same network parameters $\mathbf{w}$, we sample 100 independent mini-batches, construct the preconditioner $\mathbf{M}^{-1}$ from the first batch, and observe that after preconditioning the estimated trace of the gradient-noise covariance matrix is reduced by factors of 12x, 1505x, and 203x, respectively. This substantial reduction demonstrates that the CG–GGN preconditioner effectively attenuates gradient noise. Consistent with our theory, the combined improvements in conditioning and noise reduction yield both faster linear convergence and a significantly lower asymptotic noise floor.

## 5 Conclusion

We developed a local, geometry-aware theory for preconditioned SGD that makes two effects explicit: (1) the rate inside a basin is controlled by a preconditioner-dependent condition number in the $\mathbf{M}$–metric, and (2) the noise floor is governed by the preconditioned noise. We additionally obtained a basin-stability guarantee, giving an explicit probability that iterates remain in a region where these local properties hold. Together, the results motivate a simple rule: choose $\mathbf{M}$ to improve local conditioning while suppressing noise in the $\mathbf{M}^{-1}$–norm.

A key next direction is *covariance-aware* preconditioning. Our bounds suggest that effective design should jointly target conditioning and noise attenuation, motivating structured covariance models and adaptive schemes that update curvature and noise statistics simultaneously. Extending basin-stability guarantees

to nonstationary noise and developing online diagnostics for the local constants would move toward fully adaptive, geometry- and noise-aware SGD.

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

# Appendix

## A  Notation used in paper

In general, capital bold letters are matrices ($\mathbf{A}$), lower case bold letters are vectors ($\mathbf{v}$), and lower case Greek or Latin letters are constants ($\nu, c$). Moreover, there are some notation that is used consistently throughout the paper. A reference table for these symbols is given in Table 1.

Table 1: Reference for recurring notation in the paper.

| Symbol | Definition |
|---|---|
| $k$ | iteration counter |
| $\mathbf{w}$ | Model parameters |
| $F(\mathbf{w})$ | Objective function at point $w$ |
| $F_* := F(\mathbf{w}^*)$ | minimum function value at minimizer |
| $\alpha$ | learning rate |
| $\alpha_k$ | learning rate scheduler/ learning rate at epoch $k$ |
| $\overline{\alpha}$ | fixed learning rate |
| $c, L$ | strong convexity, Lipschitz constant for $\|\cdot\|_2 = \|\cdot\|_\mathbf{I}$ |
| $\hat{c}, \hat{L}$ | strong convexity, Lipschitz constant for preconditioned case: $\|\cdot\|_\mathbf{M}$ |
| $\hat{\mu}_{PL}$ | PL constant for preconditioned case: $\|\cdot\|_\mathbf{M}$ |
| $\mathcal{B}$ | mini-batch of the dataset |
| $\mathbf{M}$ | generic preconditioner where $\mathbf{M}^{-1}$ is applied to a vector |
| $g(\cdot, \cdot)$ | gradient vector |
| $\kappa(\mathbf{M})$ | Condition number of $\mathbf{M}$ (always based on $\|\cdot\|_2$) |
| $\mu, \mu_G$ | lower and upper bound constants on the first moment of the gradient |
| $K, K_V$ | constant and scaling values of the affine bound on the gradient's variance |
| $K_G$ | Constant needed for learning rate upper bound, dependent on $K_V + \mu_G^2 > 0$. |
| $\mathbb{E}_{\boldsymbol{\xi}}, \mathbb{V}_{\boldsymbol{\xi}}$ | Expectation and Variance of gradient with random realization $\boldsymbol{\xi}$ |
| $\beta, \gamma$ | constants affecting the lower and upper bound on $\alpha_k$ for diminishing learning rate proofs |
| $\nu$ | convergence constant in $\mathcal{O}\big((\gamma + k)^{-1}\big)$ |
| $r$ | radius of convex basin around local minimum |
| $\mathcal{N}_r, \mathcal{N}_{r_+}$ | local neighborhood around minimizer, slightly larger local neighborhood for containment |
| $\tau$ | smallest iteration number where $\mathbf{w}_k \notin \mathcal{N}_r$. |
| $C$ | The stochastic noise floor defined $\overline{\alpha}\hat{L}K/(2\hat{c}\mu)$ |
| $\mathcal{N}_\mathbf{M}(\mathbf{w})$ | instantaneous preconditioned noise $\text{tr}(\mathbf{M}^{-1}\Sigma(\mathbf{w}))$ |
| $K$ | uniform baseline for $\mathcal{N}_\mathbf{M}(\mathbf{w})$ on the analysis region (noise floor constant) |
| $\alpha_{\text{QG}}$ | quadratic growth constant of locally convex basin a distance from the minimizer |

## B  Mathematical preliminaries

### B.1  Preconditioning

The condition number from a linear equation $\mathbf{Ax} = \mathbf{b}$ bounds the accuracy of the solution $\mathbf{x}$, and is defined as

$$\kappa(\mathbf{A}) = \|\mathbf{A}\|\|\mathbf{A}^{-1}\|,$$

where if not stated $\|\cdot\| = \|\cdot\|_2$. If $\mathbf{A}$ is ill-conditioned, i.e. has a large condition number, then a small perturbation in $\mathbf{b}$ can result in a large perturbation of the solution $\mathbf{x}$. In addition to the accuracy of the solution, the convergence rate of iterative methods, such as conjugate gradient, depends on $r = \frac{\sqrt{\kappa}-1}{\sqrt{\kappa}+1}$.

It is easy to see that $r < 1$, but if $\kappa \gg 1$, then convergence will be extremely slow as $r \to 1$. This motivates the need for ways to reduce the condition number, through a technique called *preconditioning*. Throughout

this paper, we assume that $\mathbf{M}$ is the preconditioner, and we only have access to the action of $\mathbf{M}^{-1}$ onto a vector. More technically, we say $\mathbf{M}$ is an efficient preconditioner to the matrix $\mathbf{A}$ such that

$$\kappa(\mathbf{M}^{-1}\mathbf{A}) < \kappa(\mathbf{A}).$$

For clarity, even though we call $\mathbf{M}$ the preconditioner, we don't explicitly form it. Additionally, we don't form $\mathbf{M}^{-1}$ either but just observe the action of the preconditioner on a vector, $\mathbf{M}^{-1}\mathbf{v}$.

There are different ways we can utilize the preconditioner $\mathbf{M}$. First, assume $\mathbf{M}^{-1}$ exists, then the *left* preconditioned system is

$$\mathbf{M}^{-1}\left(\mathbf{A}\mathbf{x} - \mathbf{b}\right) = 0.$$

Both the original linear system and the left-preconditioned system give the same solution. Additionally, we could solve the right preconditioned system

$$\mathbf{A}\mathbf{M}^{-1}\left(\mathbf{M}\mathbf{x}\right) = \mathbf{b}.$$

This requires us to solve $\mathbf{A}\mathbf{M}^{-1}\mathbf{y} = \mathbf{b}$ for $\mathbf{y}$, and then to recover the original solution, we would need to do another linear system solve $\mathbf{M}\mathbf{x} = \mathbf{y}$ for $\mathbf{x}$.

These two techniques can be combined to perform *split* preconditioning. If we employ $\mathbf{M}$ as the right preconditioner, and $\mathbf{N}$ as the left preconditioner, we compute

$$\mathbf{N}\mathbf{A}\mathbf{M}^{-1}\left(\mathbf{M}\mathbf{x}\right) = \mathbf{N}\mathbf{b}.$$

This is beneficial if one would like to scale the rows and columns of $\mathbf{A}$ differently. Additionally, observe that if $\mathbf{A}$ is symmetric and $\mathbf{N}^{\top} = \mathbf{M}^{-1}$, then $\mathbf{N}\mathbf{A}\mathbf{M}^{-1}$ is also symmetric.

In the preconditioned version of CG (PCG), one solves the equivalent system $\mathbf{M}^{-1}\mathbf{A}\mathbf{x} = \mathbf{M}^{-1}\mathbf{b}$ using a similar three-term recurrence, but applied to the transformed system. The key requirement is that the preconditioner $\mathbf{M}$ be symmetric positive definite and chosen so that $\mathbf{M}^{-1}\mathbf{A}$ has a significantly smaller condition number than $\mathbf{A}$ itself. For practical purposes, PCG is used in matrix-free settings where only the action $\mathbf{M}^{-1}\mathbf{v}$ is required, not the explicit matrix $\mathbf{M}^{-1}$.

### B.2 Preconditioners for SGD

In this section, we briefly review several preconditioners commonly used in the ML literature. First, if we define $\mathbf{g}_k$ to be the sum of the squared gradients up until iteration $k$, we arrive at AdaGrad (Duchi et al., 2011)

$$\mathbf{M}_{\text{AdaGrad}} = \operatorname{diag}\left(\sqrt{\mathbf{g}_k} + \varepsilon\right).$$

The issues with this is the gradient squared will only increase, leading to premature stopping. To counteract that, exponentially moving weighted averages are widely used in diagonal preconditioners such as Adam (Kingma & Ba, 2017) and its momentum-less counterpart RMSProp (Hinton, 2014):

$$\mathbf{M}_{\text{Adam}} = \operatorname{diag}\left(\sqrt{\mathbf{s}_k} + \varepsilon\right),$$

where here $\mathbf{s}_k$ is an exponential moving average of squared gradients, and $\varepsilon > 0$ is a small constant added for numerical stability. While computationally efficient and robust to scaling, such diagonal preconditioners fail to capture cross-parameter curvature, which may lead to suboptimal convergence in ill-conditioned problems.

The Hessian matrix of the loss function,

$$\mathbf{H}(\mathbf{w}) = \nabla^2 \mathcal{L}(\mathbf{w}),$$

captures the exact second-order structure of the problem and provides the most complete curvature information. However, computing or storing the full Hessian is typically infeasible in high-dimensional neural network (NN) models. Moreover, it is not guaranteed to be positive definite in nonconvex settings, which complicates its direct use as a preconditioner.

To reduce computational cost, one can approximate the Hessian using a single mini-batch, $\mathcal{B}$:

$$\mathbf{H}_{\mathcal{B}}(\mathbf{w}) = \nabla^2 \mathcal{L}_{\mathcal{B}}(\mathbf{w}).$$

This matrix is cheaper to compute and can be updated online, but suffers from high variance and may not preserve important curvature directions observed over the full dataset. While the Newton and quasi-Newton methods work well for deterministic optimization, many have provided a distinction between these and other methods for designing preconditioners in the stochastic setting (Li, 2018; Bottou et al., 2018).

As opposed to constructing the Hessian, an alternative is the Gauss-Newton Hessian approximation, which assumes the difference between the model and label is small in a least-squares norm. This idea was further generalized to loss functions of the form $\ell(\theta) = \sum_n a_n (b_n (\theta))$ in Schraudolph (2002). This generalized Gauss-Newton matrix (GGN), which ignores second order information of $b_n$, is SPD when $a_n$ is convex even when the true Hessian is indefinite.

Another alternate method is the FIM defined as

$$\mathbf{F}(\mathbf{w}) = \mathbb{E}_{x,y} \left[ \nabla_{\mathbf{w}} \log p_{\mathbf{w}}(y \mid x) \nabla_{\mathbf{w}} \log p_{\mathbf{w}}(y \mid x)^{\top} \right],$$

which is guaranteed to be SPD under mild regularity conditions. For models trained with exponential-family losses, the FIM coincides with the GGN (Martens, 2020; Schraudolph, 2002). Its structure allows for stable and curvature-aware preconditioning. Using the FIM as the preconditioner in the stochastic gradient descent algorithm yields Natural Gradient Descent from online learning Amari (1998).

The empirical FIM estimates the expectation in the FIM using a finite mini-batch:

$$\mathbf{F}_{\mathrm{emp}}(\mathbf{w}) = \frac{1}{|\mathcal{B}|} \sum_{(x,y) \in \mathcal{B}} \nabla_{\mathbf{w}} \log p_{\mathbf{w}}(y \mid x) \nabla_{\mathbf{w}} \log p_{\mathbf{w}}(y \mid x)^{\top}.$$

It is symmetric and positive semidefinite, and is often used in practice due to its lower computational overhead compared to the full FIM. However, it may introduce bias depending on the mini-batch size and model quality (Kunstner et al., 2019).

Finally, the L-BFGS algorithm is a popular quasi-Newton method that builds a low-rank approximation to the inverse Hessian using a history of gradients and iterates. It is well-suited to medium-scale problems and has seen empirical success in ML (Bottou et al., 2018). Additional variants of L-BFGS have also been proposed (Berahas et al., 2016; Bollapragada et al., 2018). While not traditionally framed as a preconditioner, L-BFGS can be interpreted as implicitly applying a data-driven curvature approximation.

## C   Assumptions and proofs of theorems

### C.1   Assumptions

**Assumption 9** (Strong Convexity)**.** *The objective function $F \colon \mathbb{R}^d \to \mathbb{R}$ is strongly convex in that there exists a constant $c > 0$ such that*

$$F(\overline{\mathbf{w}}) \geq F(\mathbf{w}) + \nabla F(\mathbf{w})^{\top}(\overline{\mathbf{w}} - \mathbf{w}) + \frac{1}{2}c||\overline{\mathbf{w}} - \mathbf{w}||_2^2, \qquad \forall \, (\overline{\mathbf{w}}, \mathbf{w}) \in \mathbb{R}^d \times \mathbb{R}^d$$

From elementary optimization, this assumption is equivalent to $F$ having a unique minimizer $\mathbf{w}^* \in \mathbb{R}^d$. We define $F_* := F(\mathbf{w}^*)$.

**Assumption 10** (Lipschitz continuity of gradient)**.** *The objective function $F \colon \mathbb{R}^d \to \mathbb{R}$ is continuously differentiable and the gradient function of $F$, $\nabla F \colon \mathbb{R}^d \to \mathbb{R}^d$, is Lipschitz continuous with Lipschitz constant $L > 0$, i.e.*

$$||\nabla F(\mathbf{w}) - \nabla F(\overline{\mathbf{w}})||_2 \leq L||\mathbf{w} - \overline{\mathbf{w}}||_2$$

*for all $\{\mathbf{w}, \overline{\mathbf{w}}\} \subset \mathbb{R}^d$.*

**Remark 1.** *If $F$ is continuously twice differentiable, then $\nabla F$ is Lipschitz continuous with Lipschitz constant $L$ if and only if the eigenvalues of the matrix $\nabla^2 F(\mathbf{w})$ are bounded above by $L$ for all $w$. $F$ is strongly convex with constant $c$ if and only if the eigenvalues of the matrix $\nabla^2 F(\mathbf{w})$ is bounded below by $c$ for all $w$. Therefore, $L/c$ is an upper bound of the condition number of $\nabla^2 F(\mathbf{w})$.*

Lipschitz continuity of gradient is an assumption made in nearly all convergence analyses of gradient-based methods (Khaled & Richtárik, 2023).

**Assumption 11** (Bounds on First and Second Moments of Gradient). *Assume*

*1. There exist scalars $\mu_G \geq \mu > 0$ such that, for all $k \in \mathbb{N}$,*

$$\nabla F(\mathbf{w}_k)^\top \mathbb{E}_{\boldsymbol{\xi}_k}[g(\mathbf{w}_k, \boldsymbol{\xi}_k)] \geq \mu ||\nabla F(\mathbf{w}_k)||_2^2 \tag{9}$$

$$||\mathbb{E}_{\boldsymbol{\xi}_k}[g(\mathbf{w}_k, \boldsymbol{\xi}_k)]||_2 \leq \mu_G ||\nabla F(\mathbf{w}_k)||_2 \tag{10}$$

*2. There exist scalars $K \geq 0$ and $K_V \geq 0$ such that, for all $k \in \mathbb{N}$,*

$$\mathbb{V}_{\boldsymbol{\xi}_k}[g(\mathbf{w}_k, \boldsymbol{\xi}_k)] \leq K + K_V ||\nabla F(\mathbf{w}_k)||_2^2 \tag{11}$$

*where $\mathbb{V}_{\boldsymbol{\xi}_k}[g(\mathbf{w}_k, \boldsymbol{\xi}_k)] := \mathbb{E}_{\boldsymbol{\xi}_k}[||g(\mathbf{w}_k, \boldsymbol{\xi}_k)||_2^2] - ||\mathbb{E}_{\boldsymbol{\xi}_k}[g(\mathbf{w}_k, \boldsymbol{\xi}_k)]||_2^2$.*

**Theorem C.1** (Strongly convex objective function, fixed learning rate (Bottou et al., 2018)). *Under Assumptions 9,10, 11, suppose that the SGD algorithm is run with fixed learning rates, $\alpha_k = \overline{\alpha}$ for all $k \in \mathbb{N}$ where*

$$0 < \overline{\alpha} \leq \frac{\mu}{LK_G} \quad and \quad K_G := K_V + \mu_G^2 \geq \mu^2 > 0.$$

*Then, the expected optimality gap satisfies the following for all $k \in \mathbb{N}$:*

$$\mathbb{E}[F(\mathbf{w}_k) - F_*] \leq \frac{\overline{\alpha}LK}{2c\mu} + (1 - \overline{\alpha}c\mu)^{k-1} \left( F(\mathbf{w}_1) - F_* - \frac{\overline{\alpha}LK}{2c\mu} \right) \xrightarrow{k \to \infty} \frac{\overline{\alpha}LK}{2c\mu} \tag{12}$$

Note that it follows from (10) and (11) that $\mathbb{E}_{\boldsymbol{\xi}_k}[||g(\mathbf{w}_k, \boldsymbol{\xi}_k)||_2^2] \leq K + K_G||\nabla F(\mathbf{w}_k)||_2^2$ with $K_G := K_V + \mu_G^2 \geq \mu^2 > 0$.

**Theorem C.2** (Strongly convex objective function, diminishing learning rates (Bottou et al., 2018)). *Under the same assumptions as Theorem C.1, suppose that the SGD algorithm is run with a learning rate sequence such that, for all $k \in \mathbb{N}$,*

$$\alpha_k = \frac{\beta}{\gamma + k} \text{ for some } \beta > \frac{1}{c\mu} \text{ and } \gamma > 0 \text{ such that } \alpha_1 \leq \frac{\mu}{LK_G}$$

*Then, the expected optimality gap satisfies the following for all $k \in \mathbb{N}$:*

$$\mathbb{E}[F(\mathbf{w}_k) - F_*] \leq \frac{\nu}{\gamma + k} \tag{13}$$

*where*

$$\nu := \max \left\{ \frac{\beta^2 LK}{2(\beta c\mu - 1)}, (\gamma + 1)(F(\mathbf{w}_1) - F_*) \right\} \tag{14}$$

Under the assumption of strong convexity, the optimality gap can be bounded at any point by the 2-norm squared of the gradient of the objective function at that particular point. That is,

$$2c(F(\mathbf{w}) - F_*) \leq ||\nabla F(\mathbf{w})||_2^2 \text{ for all } \mathbf{w} \in \mathbb{R}^d$$

As before, $F$ has a unique minimizer, denoted as $\mathbf{w}^* \in \mathbb{R}^d$ with $F_* := F(\mathbf{w}^*)$.

Previously, the optimality gap was bounded by the 2-norm of the gradient of the objective function squared. Here, however, the optimality gap is bounded by the $\mathbf{M}$-norm of the gradient of the objective function squared. That is,

$$2\hat{c}(F(\mathbf{w}) - F(\mathbf{w}_*)) \leq ||\nabla F(\mathbf{w})||^2_{\mathbf{M}^{-1}}$$

This result is used several times in the upcoming proofs. We repeat Lemma 3.1 here for convenience below:

**Lemma C.3.** *Let $F$ be twice differentiable and $\mathbf{M}^{-1} = \mathbf{PP}^\top$. Then: (i) $\nabla F$ is $\mathbf{M}$-Lipschitz with constant $\hat{L} \iff$ all eigenvalues of $\mathbf{P}^\top \nabla^2 F(\mathbf{w})\mathbf{P}$ are $\leq \hat{L}$; (ii) $F$ is $\mathbf{M}$-strongly convex with constant $\hat{c} \iff$ all eigenvalues of $\mathbf{P}^\top \nabla^2 F(\mathbf{w})\mathbf{P}$ are $\geq \hat{c}$.*

*Proof.* We consider a change of parameter as used in preconditioning. Let $\mathbf{w} = \mathbf{Pz}$ and $\overline{\mathbf{w}} = \mathbf{P}\overline{\mathbf{z}}$. Then $\mathbf{w} - \overline{\mathbf{w}} = \mathbf{P}(\mathbf{z} - \overline{\mathbf{z}})$ which gives $\mathbf{P}^{-1}(\mathbf{w} - \overline{\mathbf{w}}) = \mathbf{z} - \overline{\mathbf{z}}$. Define $f(\mathbf{z}) = F(\mathbf{Pz})$. Then $\nabla_{\mathbf{z}} f(\mathbf{z}) = \mathbf{P}^\top \nabla_{\mathbf{w}} F(\mathbf{w})$ and $\nabla^2_{\mathbf{z}} f(z) = \mathbf{P}^\top \nabla^2_{\mathbf{w}} F(\mathbf{w})\mathbf{P}$. Hence

$$||\nabla f(\mathbf{z}) - \nabla f(\overline{\mathbf{z}})||_2 = ||\mathbf{P}^\top \nabla_{\mathbf{w}} F(\mathbf{w}) - \mathbf{P}^\top \nabla_{\mathbf{w}} F(\overline{\mathbf{w}})||_2 = ||\nabla_{\mathbf{w}} F(\mathbf{w}) - \nabla_{\mathbf{w}} F(\overline{\mathbf{w}})||_{\mathbf{M}^{-1}}.$$

Therefore, the $\mathbf{M}$-Lipschitz continuity of the gradient for $F$ is equivalent to the Lipschitz continuity of the gradient for $f$, which is equivalent to that $\nabla^2_z f(z)$, i.e. $\mathbf{P}^\top \nabla^2_w F(\mathbf{w})\mathbf{P}$, has eigenvalues bounded above by $\hat{L}$. Similarly, the statement on M-strong convexity follows from

$$F(\mathbf{w}) + \nabla F(\mathbf{w})^\top(\overline{\mathbf{w}} - \mathbf{w}) + \frac{1}{2}\hat{c}||\overline{\mathbf{w}} - \mathbf{w}||^2_{\mathbf{M}} = f(\mathbf{z}) + \nabla_{\mathbf{z}} f(\mathbf{z})^\top(\overline{\mathbf{z}} - \mathbf{z}) + \frac{1}{2}\hat{c}||\overline{\mathbf{z}} - \mathbf{z}||^2_2.$$

$\square$

We may assume $\hat{L}$ and $\hat{c}$ are respectively the maximum and the minimum of the eigenvalues of $\mathbf{P}^\top \nabla^2 F(\mathbf{w})\mathbf{P}$ for all $\mathbf{w}$. So $\frac{\hat{L}}{\hat{c}}$ plays the role of the condition number of the preconditioned matrix $\mathbf{P}^\top \nabla^2 F(\mathbf{w})\mathbf{P}$. If we assume $\mathbf{M}^{-1} = \mathbf{PP}^\top$ is such that $\frac{\hat{L}}{\hat{c}}$ is smaller than $\frac{L}{c}$, it basically reduces the condition number. We will demonstrate that this accelerates the speed of convergence.

An important lemma comes directly from this assumption.

**Lemma C.4.** *Under the assumption of $\mathbf{M}$-Lipschitz continuity of gradient,*

$$F(\mathbf{w}) \leq F(\overline{\mathbf{w}}) + \nabla F(\overline{\mathbf{w}})^\top(\mathbf{w} - \overline{\mathbf{w}}) + \frac{1}{2}\hat{L}||\mathbf{w} - \overline{\mathbf{w}}||^2_{\mathbf{M}} \tag{15}$$

*Proof.* Consider the following,

$$\begin{aligned}
F(\mathbf{w}) &= F(\overline{\mathbf{w}}) + \int_0^1 \left(\nabla F(\overline{\mathbf{w}} + t(\mathbf{w} - \overline{\mathbf{w}}))\right)^\top \mathbf{PP}^{-1}(\mathbf{w} - \overline{\mathbf{w}})\, \mathrm{d}t \\
&= F(\overline{\mathbf{w}}) + \nabla F(\overline{\mathbf{w}})^\top(\mathbf{w} - \overline{\mathbf{w}}) + \int_0^1 \left(\nabla F(\overline{\mathbf{w}} + t(\mathbf{w} - \overline{\mathbf{w}})) - \nabla F(\overline{\mathbf{w}})\right)^\top \mathbf{PP}^{-1}(\mathbf{w} - \overline{\mathbf{w}})\, \mathrm{d}t \\
&\leq F(\overline{\mathbf{w}}) + \nabla F(\overline{\mathbf{w}})^\top(\mathbf{w} - \overline{\mathbf{w}}) + \int_0^1 \hat{L}||t(\mathbf{w} - \overline{\mathbf{w}})||_{\mathbf{M}}||\mathbf{w} - \overline{\mathbf{w}}||_{\mathbf{M}}\, \mathrm{d}t
\end{aligned}$$

which gives us our consequence that was to be shown. $\square$

Notice that combining the variance definition (Eq. 3) with Assumption 3, we have the following

$$\mathbb{E}_{\boldsymbol{\xi}_k}[||g(\mathbf{w}_k, \boldsymbol{\xi}_k)||^2_{\mathbf{M}^{-1}}] \leq K_G||\nabla F(\mathbf{w}_k)||^2_{\mathbf{M}^{-1}} + K \text{ with } K_G := K_V + \mu_G^2 \geq \mu^2 > 0 \tag{16}$$

The proof for the two theorems relies on the following lemmas.

**Lemma C.5.** *Under Assumption 1, the iterates of Eq. 2 satisfy the following inequality for all $k \in \mathbb{N}$:*

$$\mathbb{E}_{\boldsymbol{\xi}_k}[F(\mathbf{w}_{k+1})] - F(\mathbf{w}_k) \leq -\alpha_k \nabla F(\mathbf{w}_k)^\top \mathbb{E}_{\boldsymbol{\xi}_k}[g(\mathbf{w}_k, \boldsymbol{\xi}_k)] + \frac{1}{2}\alpha_k^2 \hat{L}\mathbb{E}_{\boldsymbol{\xi}_k}[||g(\mathbf{w}_k, \boldsymbol{\xi}_k)||^2_{\mathbf{M}^{-1}}] \tag{17}$$

*Proof.* Let $\mathbf{w} = \mathbf{w}_{k+1}$ and $\overline{\mathbf{w}} = \mathbf{w}_k$. Then, by Assumption 1,

$$F(\mathbf{w}_{k+1}) - F(\mathbf{w}_k) \leq \nabla F(\mathbf{w}_k)^\top (\mathbf{w}_{k+1} - \mathbf{w}_k) + \frac{1}{2}\hat{L}||\mathbf{w}_{k+1} - \mathbf{w}_k||^2_{\mathbf{M}}$$

Recalling that Eq. 2 gives $\mathbf{w}_{k+1} = \mathbf{w}_k - \alpha_k \mathbf{M}^{-1} g(\mathbf{w}_k, \boldsymbol{\xi}_k)$, we then have,

$$F(\mathbf{w}_{k+1}) - F(\mathbf{w}_k) \leq \nabla F(\mathbf{w}_k)^\top (-\alpha_k \mathbf{M}^{-1} g(\mathbf{w}_k, \boldsymbol{\xi}_k)) + \frac{1}{2}\hat{L}|| - \alpha_k \mathbf{M}^{-1} g(\mathbf{w}_k, \boldsymbol{\xi}_k)||^2_{\mathbf{M}}$$

$$\leq -\alpha_k \nabla F(\mathbf{w}_k)^\top \mathbf{M}^{-1} g(\mathbf{w}_k, \boldsymbol{\xi}_k) + \frac{1}{2}\alpha_k^2 \hat{L}||\mathbf{M}^{-1} g(\mathbf{w}_k, \boldsymbol{\xi}_k)||^2_{\mathbf{M}}$$

$$\leq -\alpha_k \nabla F(\mathbf{w}_k)^\top \mathbf{M}^{-1} g(\mathbf{w}_k, \boldsymbol{\xi}_k) + \frac{1}{2}\alpha_k^2 \hat{L}||g(\mathbf{w}_k, \boldsymbol{\xi}_k)||^2_{\mathbf{M}^{-1}}$$

Take the expectation of both sides

$$\mathbb{E}_{\boldsymbol{\xi}_k}[F(\mathbf{w}_{k+1})] - F(\mathbf{w}_k) \leq -\alpha_k \nabla F(\mathbf{w}_k)^\top \mathbf{M}^{-1} \mathbb{E}_{\boldsymbol{\xi}_k}[g(\mathbf{w}_k, \boldsymbol{\xi}_k)] + \frac{1}{2}\alpha_k^2 \hat{L}\mathbb{E}_{\boldsymbol{\xi}_k}[||g(\mathbf{w}_k, \boldsymbol{\xi}_k)||^2_{\mathbf{M}^{-1}}]$$

Thus, the desired result is achieved. $\qquad\square$

**Lemma C.6.** *Under Assumptions 1 and 2, the iterates of Eq. 2 satisfy the following inequalities for all $k \in \mathbb{N}$:*

$$\mathbb{E}_{\boldsymbol{\xi}_k}[F(\mathbf{w}_{k+1})] - F(\mathbf{w}_k) \leq -\mu\alpha_k||\nabla F(\mathbf{w}_k)||^2_{\mathbf{M}^{-1}} + \frac{1}{2}\alpha_k^2 \hat{L}\mathbb{E}_{\boldsymbol{\xi}_k}\left[||g(\mathbf{w}_k, \boldsymbol{\xi}_k)||^2_{\mathbf{M}^{-1}}\right] \tag{18}$$

$$\leq -(\mu - \frac{1}{2}\alpha_k \hat{L}K_G)\alpha_k||\nabla F(\mathbf{w}_k)||^2_{\mathbf{M}^{-1}} + \frac{1}{2}\alpha_k^2 \hat{L}K \tag{19}$$

*Proof.* By Lemma C.5 and Assumption 2, it follows that

$$\mathbb{E}_{\boldsymbol{\xi}_k}[F(\mathbf{w}_{k+1})] - F(\mathbf{w}_k) \leq -\alpha_k\mu||\nabla F(\mathbf{w}_k)||^2_{\mathbf{M}^{-1}} + \frac{1}{2}\alpha_k^2 \hat{L}\mathbb{E}_{\boldsymbol{\xi}_k}\left[||g(\mathbf{w}_k, \boldsymbol{\xi}_k)||^2_{\mathbf{M}^{-1}}\right]$$

$$\leq -\alpha_k\mu||\nabla F(\mathbf{w}_k)||^2_{\mathbf{M}^{-1}} + \frac{1}{2}\alpha_k^2 \hat{L}\left(K_G||\nabla F(\mathbf{w}_k)||^2_{\mathbf{M}^{-1}} + K\right)$$

$$\leq -\left(\mu - \frac{1}{2}\alpha_k \hat{L}K_G\right)\alpha_k||\nabla F(\mathbf{w}_k)||^2_{\mathbf{M}^{-1}} + \frac{1}{2}\alpha_k^2 \hat{L}K$$

Hence, we have the desired inequalities. $\qquad\square$

The final lemma necessary is as follows.

**Lemma C.7.** *Under assumptions 1, 2, and 3 (with $F_*$ being the minimum of $F$), suppose Eq. 2 is run with a learning rate sequence such that for all $k \in \mathbb{N}$, assume $\alpha_k \leq \frac{\mu}{\hat{L}K_G}$. (Note that $\alpha_k$ could be constant for all $k \in \mathbb{N}$). Then the following inequality holds*

$$\mathbb{E}[F(\mathbf{w}_{k+1}) - F_*] \leq (1 - \alpha_k\hat{c}\mu)\mathbb{E}[F(\mathbf{w}_k) - F_*] + \frac{1}{2}\alpha_k^2 \hat{L}K \tag{20}$$

*Proof.* Given the assumptions and using Lemma C.6, we have $\mathbb{E}_{\boldsymbol{\xi}_k}[F(\mathbf{w}_{k+1})] - F(\mathbf{w}_k) \leq -\hat{c}\alpha_k\mu(F(\mathbf{w}_k) - F_*) + \frac{1}{2}\alpha_k^2 \hat{L}K$. Subtract $F_*$ from both sides and take the total expectation. We denote this total expectation as $\mathbb{E}[\cdot]$, which represents the expected value taken with respect to all random variables. That is, $\mathbb{E}[F(\mathbf{w}_k)] = \mathbb{E}_{\xi_1}\mathbb{E}_{\xi_2}\dots\mathbb{E}_{\xi_{k-1}}[F(\mathbf{w}_k)]$.

$$\mathbb{E}[\mathbb{E}_{\boldsymbol{\xi}_k}[F(\mathbf{w}_{k+1})] - F(\mathbf{w}_k) - F_*] \leq \mathbb{E}\left[-\hat{c}\alpha_k\mu(F(\mathbf{w}_k) - F_*) + \frac{1}{2}\alpha_k^2 \hat{L}K - F_*\right]$$

$$\mathbb{E}[\mathbb{E}_{\boldsymbol{\xi}_k}[F(\mathbf{w}_{k+1})] - F_*] \leq \mathbb{E}\left[-\hat{c}\alpha_k\mu(F(\mathbf{w}_k) - F_*) - F(\mathbf{w}_k) - F_*\right] + \frac{1}{2}\alpha_k^2 \hat{L}K$$

$$\leq \mathbb{E}\left[-\hat{c}\alpha_k\mu F(\mathbf{w}_k) + \hat{c}\alpha_k\mu F_* + F(\mathbf{w}_k) - F_*\right] + \frac{1}{2}\alpha_k^2 \hat{L}K$$

$$\leq (1 - \hat{c}\alpha_k\mu)\mathbb{E}[F(\mathbf{w}_k) - F_*] + \frac{1}{2}\alpha_k^2 \hat{L}K$$

696  which is our desired inequality (20).  □

## C.2  Proofs of main theorems

### C.2.1  Proof of Theorem 3.2

*Proof.* Using Lemma C.6, we have for all $k \in \mathbb{N}$:

$$
\begin{aligned}
\mathbb{E}_{\boldsymbol{\xi}_k}[F(\mathbf{w}_{k+1})] - F(\mathbf{w}_k) &\leq -(\mu - \frac{1}{2}\overline{\alpha}\hat{L}K_G)\overline{\alpha}||\nabla F(\mathbf{w}_k)||^2_{\mathbf{M}^{-1}} + \frac{1}{2}\overline{\alpha}^2\hat{L}K \\
&\leq -\left(\mu - \frac{1}{2}\left(\frac{\mu}{\hat{L}K_G}\right)\hat{L}K_G\right)\overline{\alpha}||\nabla F(\mathbf{w}_k)||^2_{\mathbf{M}^{-1}} + \frac{1}{2}\overline{\alpha}^2\hat{L}K \\
&= -\frac{1}{2}\overline{\alpha}\mu||\nabla F(\mathbf{w}_k)||^2_{\mathbf{M}^{-1}} + \frac{1}{2}\overline{\alpha}^2\hat{L}K \\
&\leq -\frac{1}{2}\overline{\alpha}\mu[2\hat{c}(F(\mathbf{w}_k) - F(\mathbf{w}_*))] + \frac{1}{2}\overline{\alpha}^2\hat{L}K \\
&\leq -\overline{\alpha}\hat{c}\mu(F(\mathbf{w}_k) - F_*) + \frac{1}{2}\overline{\alpha}^2\hat{L}K
\end{aligned}
$$

Now, subtract the constant $\frac{\overline{\alpha}\hat{L}K}{2\hat{c}\mu}$ from both sides of inequality (Eq. 20)

$$
\mathbb{E}[F(\mathbf{w}_{k+1}) - F_*] - \frac{\overline{\alpha}\hat{L}K}{2\hat{c}\mu} \leq (1 - \overline{\alpha}\hat{c}\mu)\mathbb{E}[F(\mathbf{w}_k) - F_*] + \frac{1}{2}\overline{\alpha}\hat{L}K - \frac{\overline{\alpha}\hat{L}K}{2\hat{c}\mu} \tag{21}
$$

$$
= (1 - \overline{\alpha}\hat{c}\mu)\left(\mathbb{E}[F(\mathbf{w}_k) - F_*] - \frac{\overline{\alpha}\hat{L}K}{2\hat{c}\mu}\right) \tag{22}
$$

699  We must now notice the following chain of inequalities.

$$
0 < \overline{\alpha}\hat{c}\mu \leq \frac{\hat{c}\mu^2}{\hat{L}K_G}
$$

700  This inequality holds by the theorem assumption that $0 < \overline{\alpha} \leq \frac{\mu}{\hat{L}K_G}$.

$$
\frac{\hat{c}\mu^2}{\hat{L}K_G} \leq \frac{\hat{c}\mu^2}{\hat{L}\mu^2} = \frac{\hat{c}}{\hat{L}}
$$

701  This inequality holds by (16) from Assumption 3.

702  Now, note that since $\hat{c} \leq \hat{L}$, it follows that $\frac{\hat{c}}{\hat{L}} \leq 1$. The result thus follows by applying $C.6$ repeatedly through
703  iteration $k \in \mathbb{N}$.  □

**Corollary C.7.1.** *If $g(\mathbf{w}_k, \boldsymbol{\xi}_k)$ is an unbiased estimate of $\nabla F(w_k)$, and the variance of $g(\mathbf{w}_k, \boldsymbol{\xi}_k)$ is bounded by a constant $K$ independent of $\nabla F(\mathbf{w}_k)$, Then for a fixed learning rate bounded by $\frac{K_G}{\hat{L}K_G}$, $\mathbb{E}[F(\mathbf{w}_k) - F_*]$ decreases to below $\frac{\overline{\alpha}\hat{L}K}{2\hat{c}\mu}$ at the rate of $\frac{\hat{c}}{\hat{L}}$.*

### C.2.2  Proof of Theorem 3.3

*Proof.* Since the learning rates are diminishing and by the theorem statement, we have $\alpha_k\hat{L}K_G \leq \alpha_1\hat{L}K_G \leq \mu$ for all $k \in \mathbb{N}$. By Lemma C.6 and Assumption 3,

$$
\begin{aligned}
\mathbb{E}_{\boldsymbol{\xi}_k}[F(\mathbf{w}_{k+1})] - F(\mathbf{w}_k) &\leq -(\mu - \frac{1}{2}\alpha_k\hat{L}K_G)\alpha_k||\nabla F(\mathbf{w}_k)||^2_{\mathbf{M}^{-1}} + \frac{1}{2}\alpha_k^2\hat{L}K \\
&\leq -(\mu - \frac{1}{2}\mu)\alpha_k||\nabla F(\mathbf{w}_k)||^2_{\mathbf{M}^{-1}} + \frac{1}{2}\alpha_k^2\hat{L}K \\
&\leq -\alpha_k\mu\hat{c}(F(\mathbf{w}_k) - F_*) + \frac{1}{2}\alpha_k^2\hat{L}K
\end{aligned}
$$

By Lemma C.7, using (20), we have

$$\mathbb{E}[F(\mathbf{w}_{k+1}) - F_*] \leq (1 - \alpha_k \hat{c}\mu)\mathbb{E}[F(\mathbf{w}_k) - F_*] + \frac{1}{2}\alpha_k^2 \hat{L}K$$

Now, we prove the convergence result via induction. Consider the base case, $k = 1$.

Since $\nu \geq (\gamma + 1)(F(\mathbf{w}_1) - F_*)$ and $\nu \geq \frac{\beta^2 \hat{L}K}{2(\beta \hat{c}\mu - 1)}$, it follows that $\mathbb{E}[F(\mathbf{w}_1) - F_*] \leq \frac{\nu}{\gamma + 1}$.

Now, we assume that (8) holds for some $k \geq 1$. Thus

$$\begin{aligned}
\mathbb{E}[F(\mathbf{w}_{k+1}) - F_*] &\leq (1 - \alpha_k \hat{c}\mu)\mathbb{E}[F(\mathbf{w}_k) - F_*] + \frac{1}{2}\alpha_k^2 \hat{L}K \\
&\leq (1 - \alpha_k \hat{c}\mu)\frac{\nu}{\gamma + k} + \frac{1}{2}\alpha_k^2 \hat{L}K \\
&= \left(1 - \frac{\beta}{\gamma + k}\hat{c}\mu\right)\frac{\nu}{\gamma + k} + \frac{1}{2}\left(\frac{\beta}{\gamma + k}\right)^2 \hat{L}K \\
&= \left(1 - \frac{\beta\hat{c}\mu}{\tilde{k}}\right)\frac{\nu}{\tilde{k}} + \frac{\beta^2 \hat{L}K}{2\tilde{k}^2} \\
&= \left(\frac{\tilde{k} - 1}{\tilde{k}^2}\right)\nu - \left(\frac{\beta\hat{c}\mu - 1}{\tilde{k}^2}\right)\nu + \frac{\beta^2 \hat{L}K}{2\tilde{k}^2}
\end{aligned}$$

where $\tilde{k} := \gamma + k$. Note that $\left(\frac{\beta\hat{c}\mu - 1}{\tilde{k}^2}\right)\nu - \frac{\beta^2 \hat{L}K}{2\tilde{k}^2} \geq 0$ since $\nu \geq \frac{\beta^2 \hat{L}K}{2(\beta \hat{c}\mu - 1)}$.

Thus,

$$\mathbb{E}[F(\mathbf{w}_{k+1}) - F_*] \leq \left(\frac{\tilde{k} - 1}{\tilde{k}^2}\right)\nu - \left(\frac{\beta\hat{c}\mu - 1}{\tilde{k}^2}\right)\nu + \frac{\beta\hat{L}K}{2\tilde{k}^2} \stackrel{\dagger}{\leq} \frac{\nu}{\tilde{k} + 1}$$

where ($\dagger$) follows since $\tilde{k}^2 \geq (\tilde{k} + 1)(\tilde{k} - 1)$. $\qquad\square$

### C.2.3 Proof of Lemma 3.4

*Proof.* Fix $k \leq T - 1$ and assume $\mathbf{w}_k \in \mathcal{N}_r$, i.e. $\text{dist}_{\mathbf{M}}(\mathbf{w}_k, \mathcal{S}) \leq r$. If $\mathbf{w}_{k+1} \notin \mathcal{N}_{r_+}$ then $\text{dist}_{\mathbf{M}}(\mathbf{w}_{k+1}, \mathcal{S}) > r_+ = r + \Delta$. By the triangle inequality,

$$\text{dist}_{\mathbf{M}}(\mathbf{w}_{k+1}, \mathcal{S}) \leq \text{dist}_{\mathbf{M}}(\mathbf{w}_k, \mathcal{S}) + \|\mathbf{w}_{k+1} - \mathbf{w}_k\|_{\mathbf{M}} \leq r + \|\mathbf{w}_{k+1} - \mathbf{w}_k\|_{\mathbf{M}},$$

hence $\|\mathbf{w}_{k+1} - \mathbf{w}_k\|_{\mathbf{M}} > \Delta$. Using $\mathbf{w}_{k+1} - \mathbf{w}_k = -\alpha_k \mathbf{M}^{-1}g_k$ we have $\|\mathbf{w}_{k+1} - \mathbf{w}_k\|_{\mathbf{M}} = \alpha_k\|g_k\|_{\mathbf{M}^{-1}}$, so

$$\mathbb{P}(\mathbf{w}_{k+1} \notin \mathcal{N}_{r_+} \mid \mathcal{F}_k) \leq \mathbb{P}(\alpha_k\|g_k\|_{\mathbf{M}^{-1}} > \Delta \mid \mathcal{F}_k).$$

Markov's inequality and Assumption 8 yield

$$\mathbb{P}(\alpha_k\|g_k\|_{\mathbf{M}^{-1}} > \Delta \mid \mathcal{F}_k) \leq \frac{\alpha_k^2 \,\mathbb{E}[\|g_k\|_{\mathbf{M}^{-1}}^2 \mid \mathcal{F}_k]}{\Delta^2} \leq \delta_k.$$

$\qquad\square$

### C.2.4 Proof of Theorem 3.5

*Proof.* Fix $\alpha_k = \overline{\alpha}$ and let $\mathcal{F}_k := \sigma(\boldsymbol{\xi}_1, \ldots, \boldsymbol{\xi}_{k-1})$. Write $g_k := g(\mathbf{w}_k, \boldsymbol{\xi}_k)$ and define

$$\tau := \inf\{k \geq 1 : \mathbf{w}_k \notin \mathcal{N}_r\}, \qquad \Omega_T := \{\tau > T\}.$$

Fix $k \leq T - 1$ and work on $\Omega_T$. Then $\mathbf{w}_k, \mathbf{w}_{k+1} \in \mathcal{N}_r \subset \mathcal{N}_{r_+} \subset \mathcal{V}$. By convexity of $\mathcal{V}$, the segment $[\mathbf{w}_k, \mathbf{w}_{k+1}] \subset \mathcal{V}$, and by Assumption 5 (local $\mathbf{M}$–smoothness),

$$F(\mathbf{w}_{k+1}) \leq F(\mathbf{w}_k) - \overline{\alpha}\,\nabla F(\mathbf{w}_k)^\top \mathbf{M}^{-1}g_k + \frac{\hat{L}}{2}\overline{\alpha}^2\|g_k\|_{\mathbf{M}^{-1}}^2 \qquad \text{on } \Omega_T. \tag{23}$$

Taking conditional expectation given $(\mathcal{F}_k, \Omega_T)$ and using the conditional-moment version of Assumption 6 on $\Omega_T$ yields

$$\mathbb{E}[F(\mathbf{w}_{k+1}) - F_* \mid \mathcal{F}_k, \Omega_T] \leq (F(\mathbf{w}_k) - F_*) - \overline{\alpha}\,\mu\,\|\nabla F(\mathbf{w}_k)\|_{\mathbf{M}^{-1}}^2$$
$$+ \frac{\hat{L}}{2}\overline{\alpha}^2\Big(K_G\|\nabla F(\mathbf{w}_k)\|_{\mathbf{M}^{-1}}^2 + K\Big).$$

Using $\overline{\alpha} \leq \mu/(\hat{L}K_G)$ gives $\overline{\alpha}\mu - \frac{\hat{L}}{2}\overline{\alpha}^2 K_G \geq \frac{\mu}{2}\overline{\alpha}$, hence

$$\mathbb{E}[F(\mathbf{w}_{k+1}) - F_* \mid \mathcal{F}_k, \Omega_T] \leq (F(\mathbf{w}_k) - F_*) - \frac{\mu}{2}\overline{\alpha}\,\|\nabla F(\mathbf{w}_k)\|_{\mathbf{M}^{-1}}^2 + \frac{\hat{L}}{2}\overline{\alpha}^2 K. \tag{24}$$

On $\Omega_T$ we have $\mathbf{w}_k \in \mathcal{N}_r$, so Assumption 4 implies $\|\nabla F(\mathbf{w}_k)\|_{\mathbf{M}^{-1}}^2 \geq 2\hat{\mu}_{\mathrm{PL}}(F(\mathbf{w}_k) - F_*)$. Substituting into (24) gives

$$\mathbb{E}[F(\mathbf{w}_{k+1}) - F_* \mid \mathcal{F}_k, \Omega_T] \leq (1-\rho)\,(F(\mathbf{w}_k) - F_*) + \rho C,$$

with $\rho := \overline{\alpha}\hat{\mu}_{\mathrm{PL}}\mu \in (0,1)$ and $C := \frac{\overline{\alpha}\hat{L}K}{2\hat{\mu}_{\mathrm{PL}}\mu}$. Taking expectations under $\mathbb{P}(\cdot \mid \Omega_T)$ and defining $x_k := \mathbb{E}[F(\mathbf{w}_k) - F_* \mid \Omega_T]$ yields for $k \leq T-1$,

$$x_{k+1} \leq (1-\rho)x_k + \rho C.$$

Iterating gives, for all $1 \leq k \leq T$,

$$x_k \leq C + (1-\rho)^{k-1}\big(F(\mathbf{w}_1) - F_* - C\big),$$

which is the desired conditional geometric bound.

Define overshoot events

$$A_k := \{\mathbf{w}_k \in \mathcal{N}_r,\ \mathbf{w}_{k+1} \notin \mathcal{N}_{r_+}\}, \qquad k = 1,\ldots,T-1,$$

and the no-overshoot event $\mathcal{E}_T := \bigcap_{k=1}^{T-1} A_k^c$. By Lemma 3.4, $\mathbb{P}(A_k) \leq \delta_k$, hence by the union bound

$$\mathbb{P}(\mathcal{E}_T^c) \leq \sum_{k=1}^{T-1} \delta_k. \tag{25}$$

Let $\sigma := \tau \wedge T$. On $\mathcal{E}_T \cap \{\tau \leq T\}$ we have $\mathbf{w}_\tau \in \mathcal{N}_{r_+} \setminus \mathcal{N}_r$, hence by Assumption 7,

$$F(\mathbf{w}_\tau) - F_* \geq B := \frac{\alpha_{\mathrm{QG}}}{2}r^2.$$

Since $\mathbf{w}_\sigma = \mathbf{w}_\tau$ on $\{\tau \leq T\}$,

$$B\,\mathbf{1}_{\{\tau \leq T\}}\mathbf{1}_{\mathcal{E}_T} \leq (F(\mathbf{w}_\sigma) - F_*)\,\mathbf{1}_{\mathcal{E}_T}.$$

Taking expectations gives

$$B\,\mathbb{P}(\tau \leq T, \mathcal{E}_T) \leq \mathbb{E}[(F(\mathbf{w}_\sigma) - F_*)\mathbf{1}_{\mathcal{E}_T}]. \tag{26}$$

We upper bound the RHS of (26). For each $k = 1,\ldots,T-1$, define the prefix no-overshoot event

$$\mathcal{E}_{k+1} := \bigcap_{j=1}^{k} A_j^c,$$

so that $\mathcal{E}_{k+1} \in \mathcal{F}_{k+1}$ and $\mathcal{E}_T \subseteq \mathcal{E}_{k+1}$. On $\mathcal{E}_{k+1} \cap \{k < \tau\}$ we have $\mathbf{w}_k \in \mathcal{N}_r$ and $\mathbf{w}_{k+1} \in \mathcal{N}_{r_+} \subset \mathcal{V}$, so by smoothness,

$$F(\mathbf{w}_{k+1}) - F(\mathbf{w}_k) \leq -\overline{\alpha}\,\nabla F(\mathbf{w}_k)^\top \mathbf{M}^{-1}g_k + \frac{\hat{L}}{2}\overline{\alpha}^2\|g_k\|_{\mathbf{M}^{-1}}^2 \qquad \text{on } \mathcal{E}_{k+1} \cap \{k < \tau\}.$$

Taking conditional expectation given $\mathcal{F}_k$ and using Assumption 6 (valid on $\{k < \tau\}$ since then $\mathbf{w}_k \in \mathcal{N}_r$) yields

$$\mathbb{E}[F(\mathbf{w}_{k+1}) - F(\mathbf{w}_k) \mid \mathcal{F}_k] \leq -\overline{\alpha}\mu\|\nabla F(\mathbf{w}_k)\|^2_{\mathbf{M}^{-1}} + \frac{\hat{L}}{2}\overline{\alpha}^2\big(K_G\|\nabla F(\mathbf{w}_k)\|^2_{\mathbf{M}^{-1}} + K\big) \leq \frac{\hat{L}}{2}\overline{\alpha}^2 K,$$

where the last inequality uses that the first term is nonpositive and we drop it.

Now note that $F(\mathbf{w}_\sigma) - F(\mathbf{w}_1) = \sum_{k=1}^{T-1}\big(F(\mathbf{w}_{k+1}) - F(\mathbf{w}_k)\big)\mathbf{1}_{\{k<\tau\}}$ and that on $\mathcal{E}_T$ we have $\mathcal{E}_T \subseteq \mathcal{E}_{k+1}$, hence the above bound applies on $\mathcal{E}_T \cap \{k < \tau\}$ for every $k \leq T - 1$. Therefore,

$$\begin{aligned}
\mathbb{E}[(F(\mathbf{w}_\sigma) - F(\mathbf{w}_1))\mathbf{1}_{\mathcal{E}_T}] &= \sum_{k=1}^{T-1} \mathbb{E}\big[(F(\mathbf{w}_{k+1}) - F(\mathbf{w}_k))\mathbf{1}_{\mathcal{E}_T}\mathbf{1}_{\{k<\tau\}}\big] \\
&= \sum_{k=1}^{T-1} \mathbb{E}\big[\mathbf{1}_{\mathcal{E}_T}\mathbf{1}_{\{k<\tau\}}\mathbb{E}[F(\mathbf{w}_{k+1}) - F(\mathbf{w}_k) \mid \mathcal{F}_k]\big] \\
&\leq \sum_{k=1}^{T-1} \frac{\hat{L}}{2}\overline{\alpha}^2 K = \frac{\hat{L}}{2}\overline{\alpha}^2 K\,(T-1),
\end{aligned}$$

which implies

$$\mathbb{E}[(F(\mathbf{w}_\sigma) - F_*)\mathbf{1}_{\mathcal{E}_T}] \leq (F(\mathbf{w}_1) - F_*) + \frac{\hat{L}}{2}\overline{\alpha}^2 K\,(T-1). \tag{27}$$

Combining (26) and (27) yields

$$\mathbb{P}(\tau \leq T, \mathcal{E}_T) \leq \frac{F(\mathbf{w}_1) - F_* + \frac{\hat{L}}{2}\overline{\alpha}^2 K\,(T-1)}{B}.$$

Finally, using (25),

$$\mathbb{P}(\tau \leq T) \leq \mathbb{P}(\tau \leq T, \mathcal{E}_T) + \mathbb{P}(\mathcal{E}_T^c) \leq \frac{F(\mathbf{w}_1) - F_* + \frac{\hat{L}}{2}\overline{\alpha}^2 K\,(T-1)}{B} + \sum_{k=1}^{T-1}\delta_k,$$

and rearranging gives the stated lower bound on $\mathbb{P}(\tau > T)$ (with truncation at 0). $\qquad\square$

### C.2.5  Proof of Theorem 3.6

*Proof.* Let $\mathcal{F}_k := \sigma(\boldsymbol{\xi}_1, \ldots, \boldsymbol{\xi}_{k-1})$, set $\alpha_k = \beta/(\gamma + k)$, and write $g_k := g(\mathbf{w}_k, \boldsymbol{\xi}_k)$. Define $\tau := \inf\{k \geq 1 : \mathbf{w}_k \notin \mathcal{N}_r\}$, $\Omega_T := \{\tau > T\}$, and $S_k := F(\mathbf{w}_k) - F_*$.

Fix $k \leq T - 1$ and work on $\Omega_T$. Then $\mathbf{w}_k, \mathbf{w}_{k+1} \in \mathcal{N}_r \subset \mathcal{N}_{r_+} \subset \mathcal{V}$. Since $\mathcal{V}$ is convex, $[\mathbf{w}_k, \mathbf{w}_{k+1}] \subset \mathcal{V}$ and Assumption 5 implies the $\mathbf{M}$–smoothness inequality:

$$F(\mathbf{w}_{k+1}) \leq F(\mathbf{w}_k) - \alpha_k\,\nabla F(\mathbf{w}_k)^\top \mathbf{M}^{-1} g_k + \frac{\hat{L}}{2}\alpha_k^2\|g_k\|^2_{\mathbf{M}^{-1}} \qquad \text{on } \Omega_T.$$

Take conditional expectation given $\mathcal{F}_k$ and using Assumption 6 (valid on $\{k < \tau\}$ since then $\mathbf{w}_k \in \mathcal{N}_r$) yields:

$$\mathbb{E}[S_{k+1} \mid \mathcal{F}_k, \Omega_T] \leq S_k - \alpha_k\mu\|\nabla F(\mathbf{w}_k)\|^2_{\mathbf{M}^{-1}} + \frac{\hat{L}}{2}\alpha_k^2\big(K_G\|\nabla F(\mathbf{w}_k)\|^2_{\mathbf{M}^{-1}} + K\big).$$

Because $\alpha_k \leq \alpha_1 = \beta/(\gamma+1) \leq \mu/(\hat{L}K_G)$, we have $\mu\alpha_k - \frac{\hat{L}}{2}\alpha_k^2 K_G \geq \frac{\mu}{2}\alpha_k$, hence

$$\mathbb{E}[S_{k+1} \mid \mathcal{F}_k, \Omega_T] \leq S_k - \frac{\mu}{2}\alpha_k\|\nabla F(\mathbf{w}_k)\|^2_{\mathbf{M}^{-1}} + \frac{\hat{L}}{2}\alpha_k^2 K.$$

On $\Omega_T$ we have $\mathbf{w}_k \in \mathcal{N}_r$, so Assumption 4 yields $\|\nabla F(\mathbf{w}_k)\|^2_{\mathbf{M}^{-1}} \geq 2\hat{\mu}_{\mathrm{PL}}S_k$. Therefore, with $m := \mu\hat{\mu}_{\mathrm{PL}}$ and $c := \hat{L}K/2$,

$$\mathbb{E}[S_{k+1} \mid \mathcal{F}_k, \Omega_T] \leq (1 - m\alpha_k)S_k + c\alpha_k^2.$$

Now take expectation under $\mathbb{P}(\cdot \mid \Omega_T)$ and define $x_k := \mathbb{E}[S_k \mid \Omega_T]$. Then for all $k \leq T - 1$,

$$x_{k+1} \leq (1 - m\alpha_k)x_k + c\alpha_k^2.$$

Substituting $\alpha_k = \beta/(\gamma + k)$ gives

$$x_{k+1} \leq \left(1 - \frac{a}{\gamma + k}\right)x_k + \frac{b}{(\gamma + k)^2}, \qquad a := \beta m, \quad b := c\beta^2.$$

Since $\beta > 2/(\hat{\mu}_{\mathrm{PL}}\mu)$, we have $a > 1$. Let

$$\nu := \max\left\{\frac{b}{a - 1}, (\gamma + 1)x_1\right\}, \qquad x_1 = F(\mathbf{w}_1) - F_*.$$

We prove by induction that $x_k \leq \nu/(\gamma + k)$ for $1 \leq k \leq T$. The base case holds because $x_1 \leq \nu/(\gamma + 1)$ by definition of $\nu$. Assuming $x_k \leq \nu/(\gamma + k)$, we obtain

$$x_{k+1} \leq \left(1 - \frac{a}{\gamma + k}\right)\frac{\nu}{\gamma + k} + \frac{b}{(\gamma + k)^2} = \frac{\nu}{\gamma + k} + \frac{b - a\nu}{(\gamma + k)^2}.$$

Using $\nu \geq b/(a - 1)$ implies $b - a\nu \leq -\nu$, hence

$$x_{k+1} \leq \frac{\nu}{\gamma + k} - \frac{\nu}{(\gamma + k)^2} \leq \frac{\nu}{\gamma + k} - \frac{\nu}{(\gamma + k)(\gamma + k + 1)} = \frac{\nu}{\gamma + k + 1}.$$

Thus $x_k \leq \nu/(\gamma + k)$ for all $1 \leq k \leq T$, i.e.

$$\mathbb{E}[F(\mathbf{w}_k) - F_* \mid \Omega_T] \leq \frac{\nu}{\gamma + k}, \qquad 1 \leq k \leq T.$$

Define overshoot events $A_k := \{\mathbf{w}_k \in \mathcal{N}_r, \ \mathbf{w}_{k+1} \notin \mathcal{N}_{r_+}\}$ for $k = 1, \ldots, T - 1$ and $\mathcal{E}_T := \bigcap_{k=1}^{T-1} A_k^c$. By Lemma 3.4, $\mathbb{P}(A_k) \leq \delta_k$, hence

$$\mathbb{P}(\mathcal{E}_T^c) \leq \sum_{k=1}^{T-1} \delta_k.$$

Let $\sigma := \tau \wedge T$. On $\mathcal{E}_T \cap \{\tau \leq T\}$ we have $\mathbf{w}_\tau \in \mathcal{N}_{r_+} \setminus \mathcal{N}_r$, so Assumption 7 yields

$$F(\mathbf{w}_\tau) - F_* \geq B := \frac{\alpha_{\mathrm{QG}}}{2}r^2.$$

Since $\mathbf{w}_\sigma = \mathbf{w}_\tau$ on $\{\tau \leq T\}$, it follows that

$$B\,\mathbf{1}_{\{\tau \leq T\}}\mathbf{1}_{\mathcal{E}_T} \leq (F(\mathbf{w}_\sigma) - F_*)\mathbf{1}_{\mathcal{E}_T}.$$

Taking expectations gives

$$B\,\mathbb{P}(\tau \leq T, \mathcal{E}_T) \leq \mathbb{E}[(F(\mathbf{w}_\sigma) - F_*)\mathbf{1}_{\mathcal{E}_T}].$$

We upper bound the right-hand side by telescoping. For $k = 1, \ldots, T-1$, define the prefix event $\mathcal{E}_k := \bigcap_{j=1}^{k-1} A_j^c$ (so $\mathcal{E}_k \in \mathcal{F}_k$ and $\mathcal{E}_T \subseteq \mathcal{E}_k$). On $\mathcal{E}_k \cap \{k < \tau\}$ we have $\mathbf{w}_k \in \mathcal{N}_r$ and $\mathbf{w}_{k+1} \in \mathcal{N}_{r_+} \subset \mathcal{V}$, so the smoothness inequality and Assumption 6 imply

$$\mathbb{E}[F(\mathbf{w}_{k+1}) - F(\mathbf{w}_k) \mid \mathcal{F}_k] \leq c\,\alpha_k^2 \qquad \text{on } \mathcal{E}_k \cap \{k < \tau\},$$

using again $\alpha_k \leq \mu/(\hat{L}K_G)$ to drop the (nonpositive) gradient-dependent part. Multiplying by $\mathbf{1}_{\mathcal{E}_T}\mathbf{1}_{\{k<\tau\}}$ and taking expectations yields

$$\mathbb{E}[(F(\mathbf{w}_{k+1}) - F(\mathbf{w}_k))\,\mathbf{1}_{\mathcal{E}_T}\mathbf{1}_{\{k<\tau\}}] \leq c\,\alpha_k^2.$$

Summing over $k = 1, \ldots, T - 1$ and using $F(\mathbf{w}_\sigma) - F(\mathbf{w}_1) = \sum_{k=1}^{T-1}(F(\mathbf{w}_{k+1}) - F(\mathbf{w}_k))\mathbf{1}_{\{k < \tau\}}$ gives

$$\mathbb{E}[(F(\mathbf{w}_\sigma) - F(\mathbf{w}_1))\mathbf{1}_{\mathcal{E}_T}] \le c \sum_{k=1}^{T-1} \alpha_k^2,$$

hence

$$\mathbb{E}[(F(\mathbf{w}_\sigma) - F_*)\mathbf{1}_{\mathcal{E}_T}] \le (F(\mathbf{w}_1) - F_*) + c \sum_{k=1}^{T-1} \alpha_k^2.$$

Therefore,

$$\mathbb{P}(\tau \le T, \mathcal{E}_T) \le \frac{F(\mathbf{w}_1) - F_* + c \sum_{k=1}^{T-1} \alpha_k^2}{B}.$$

Finally,

$$\mathbb{P}(\tau > T) \ge 1 - \mathbb{P}(\tau \le T, \mathcal{E}_T) - \mathbb{P}(\mathcal{E}_T^c) \ge 1 - \frac{F(\mathbf{w}_1) - F_* + c \sum_{k=1}^{T-1} \alpha_k^2}{B} - \sum_{k=1}^{T-1} \delta_k,$$

and truncation gives the $\max\{0, \cdot\}$ form. $\qquad\square$

# D  Numerical experiments

## D.1  Implementation details

The algorithms in this paper were implemented in Python using `jax` (version 0.5.0), `flax` (version 0.10.0), and `optax` (version 0.2.4). All timing results reported in Section 4 were measured on a consistent hardware platform running Ubuntu 24.04.2 LTS, equipped with an Intel(R) Core(TM) i7-12700K CPU (8 Performance-cores @ 3.60 GHz and 4 Efficient-cores @ 2.70 GHz), and 64 GB of system memory. All experiments were executed in double precision arithmetic to ensure numerical stability for the challenging SciML problems.

## D.2  Baseline methods and experimental setting

Our experiments evaluated several optimization algorithms to validate our theoretical analysis of precon-ditioning effects. We implemented vanilla SGD, SGD with momentum ($\beta = 0.9$), and the preconditioned methods using GGN and Hessian approximations. The goal was to minimize the mean square error loss. For plotting purposes, we normalized the training losses so that the first loss was recorded as 1.0. The preconditioned methods employ conjugate gradient to efficiently approximate matrix-vector products with the inverse preconditioner, avoiding the prohibitive cost of explicitly forming and inverting the full matrices. This approach provides a computationally tractable way to incorporate curvature information into the optimization process. For Adam (with $\beta_1 = 0.9$, $\beta_2 = 0.999$) and L-BFGS (with memory size 100 and maximum line search of 100 steps), we utilized the implementations available in the `optax` library.

Our experimental protocol employed a structured two-phase optimization strategy. Phase I uses Adam to reach a comparable local basin; Phase II switches to the target optimizer to isolate late-stage behavior. Because our nonconvex theory is local, the basin reached at the end of Phase I can influence the local constants ($\hat{L}, \hat{\mu}_{\mathrm{PL}}, K$) encountered in Phase II and hence may affect which optimizer performs best after the switch. We therefore use the same Adam warm start, switch point, architecture, and seed protocol across all methods to control for basin selection and interpret the Phase II results as comparisons conditional on entering a comparable basin rather than fully basin-agnostic rankings. This established a common starting point in the optimization landscape and helped navigate past initial high-gradient regions. In Phase II, we transitioned to the respective optimization methods for direct performance comparison. The specific duration of each phase varied by task complexity and is detailed in the respective experimental sections.

We individually optimized learning rates for each method-task combination through grid search, deliberately omitting learning rate schedulers to isolate the inherent convergence properties of each optimizer. For Adam, we searched within the range $\{0.001, 0.0005, 0.0002, 0.0001, \ldots, 0.00001\}$. The preconditioned methods required different learning rate ranges due to their curvature properties: CG-Hessian and CG-GGN used

{1.0, 0.5, . . . , 0.001}. This difference reflects our theoretical analysis that effective preconditioning can support larger learning rates when operating near local minima. For vanilla SGD and momentum SGD, we initially explored the same ranges as Adam and expanded to wider intervals when necessary to ensure optimal performance. This methodology ensured a fair comparison by allowing each optimizer to operate at its most effective learning rate for each specific task.

To ensure robust experimental results, we conducted each experiment five times using different random seeds (42 to 46 for Phase I and 43 to 47 for Phase II). This approach accounts for the inherent stochasticity in neural network training processes and allows us to report mean performance metrics. For our timing analysis, we implemented a precise measurement protocol that isolates the computational efficiency of the optimization methods themselves. Specifically, we excluded all data generation and preprocessing overhead, capturing only the cumulative duration of the actual training iterations on identical hardware configurations. This methodology provides an equitable assessment of computational efficiency, particularly important when comparing methods with substantially different per-iteration costs, such as first-order methods versus preconditioned approaches that require conjugate gradient iterations.

### D.3   Noisy data regression

For the Franke function regression experiment, we used a neural network with two hidden layers of 50 neurons each and ReLU activation functions. We resampled the dataset every epoch, generating 256 points with additive Gaussian noise as described in Section 4.2 and illustrated in the left panel of Figure 6. For the preconditioned methods, we employed 5 conjugate gradient iterations. The right panel of Figure 6 extends our main results by displaying not only the mean performance across 5 independent runs but also the variance bands for each optimization method.

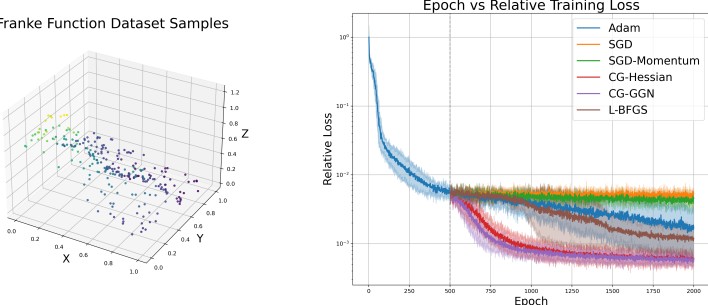

Figure 6: Left: Visualization of the Franke function dataset sampling. Right: Franke function regression performance averaged over 5 independent runs. Left: Training loss versus epochs with Phase I transitioning to Phase II at epoch 500 with variance.

### D.4   Physics-informed neural networks

For solving the Poisson equation with PINNs, we used a neural network with two hidden layers of 50 neurons each and tanh activation functions. We resampled the dataset every epoch, generating 1,000 points within the domain and 200 points on the boundary, as described in Appendix 4.2 and illustrated in the left panel of Figure 7. For the preconditioned methods, we employed 20 conjugate gradient iterations. The right panel of Figure 7 shows that the mean loss trajectory is accompanied by a tight variance envelope across 5 independent runs.

### D.5   Green's function learning

For both cases in the Green's function experiments, we used a neural network with five hidden layers of 20 neurons each and tanh activation functions. We resampled the dataset every epoch, generating 1,000 points within the domain, 500 points such that $x$ is close to $y$, and 200 points on the boundary. For the

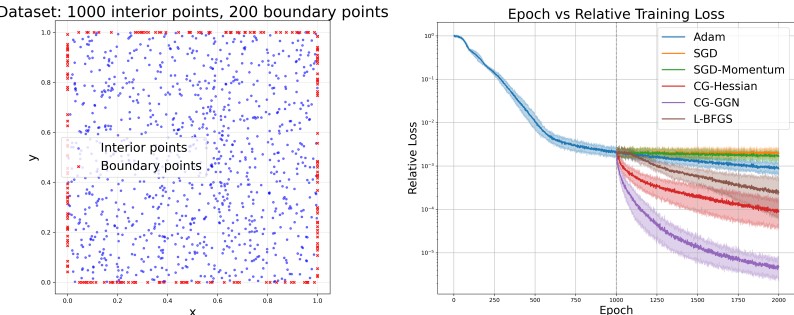

Figure 7: Left: Visualization of the sampling strategy for the 2D Poisson equation PINNs. The plot shows the distribution of $1,000$ collocation points within the domain (blue) and $200$ points along the boundary (red) used for enforcing the PDE and boundary conditions respectively. Right: Poisson equation PINNs performance averaged over 5 independent runs. Training loss versus epochs with Phase I transitioning to Phase II at epoch $1,000$ with variance.

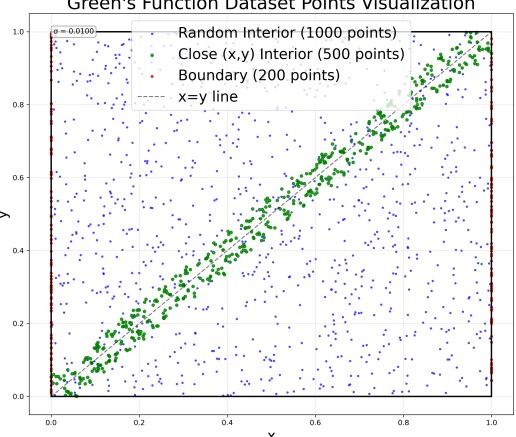

Figure 8: Visualization of the sampling strategy for Green's function learning. The plot shows three categories of training points: randomly distributed interior points (blue, $1,000$ points), points concentrated near the diagonal where $x$ is close to $y$ (green, $500$ points) to capture the near-singularity behavior characteristic of Green's functions, and boundary points (red, $200$ points) used to enforce homogeneous Dirichlet boundary conditions.

838 preconditioned methods, we employed 20 conjugate gradient iterations. Figure 9 extends our main results by
839 displaying not only the mean performance across 5 independent runs but also the variance bands for each
840 optimization method.

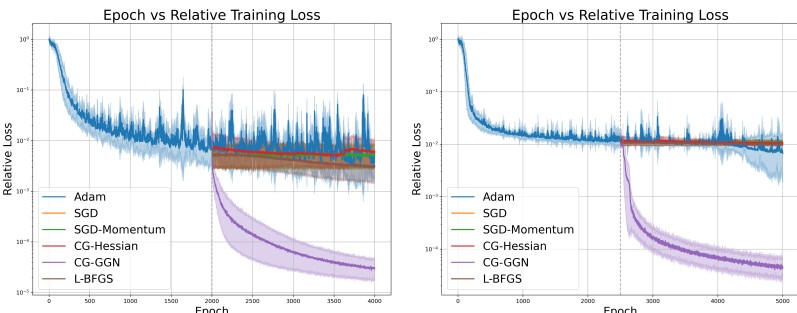

Figure 9: Green's function learning performance averaged over 5 independent runs. Left: Training loss versus epochs with Phase I transitioning to Phase II at epoch $2,000$ with variance for Laplacian. Right: Training loss versus epochs with Phase I transitioning to Phase II at epoch $2,500$ with variance for convection-diffusion.

