# OpenReview forum: "Design Criteria for SGD Preconditioners: Local Conditioning, Noise Floors, and Basin Stability"
_TMLR — Accepted by TMLR_

### Review · Reviewer_21Ha · 2025-12-18

**Summary Of Contributions:**

#### **Summary**
The authors extended the linear convergence theory of SGD in Euclidean space to the space defined by the preconditioner M.
They first showed that similar linear convergence (in a different M-induced norm) holds in the strongly convex case.
Then they extended this argument to the local non-convex but flat regime, examplifying the later stage of over-parametrized model training.
After showing these results theoretically, they tested several variants of SGD on SciML tasks.

#### **Strength**
- They theoretically analyzed the convergence of pre-conditioning method, which seems not done before (not sure though).
- I guess this is an important problem as several pre-conditioning methods are coming out.
- It seems the overall theoretical derivations are rigorous (though I could not check every line of the proof)

#### **Weakness**
- It needs a lot of modification in writing. In current form, it is very difficult to capture its own contribution and novelty. For instance,
  - The title includes 'designing preconditioner' but the paper is not about 'designing'.
  - Section 1. Introduction part begins with the enumerating SGD theories and then suddenly presents 'pre-conditioning' as its main theme. Then suddenly SciML appears. It is very difficult to capture what the paper want to present.
  - Also in 1. Introduction, existing works and the paper's contributions are mixed, so it is difficult to capture what is old and what is new. For example, the last two lines of page 1 seem like the paper's own contribution. But it is presented as it has already been known before.
  - The term 'condition number' is key throughout the paper, and appears from the first page. But it is never defined until appendix.
  - Figure 1 in the introduciton section should give a sense of 'what the paper will present.' But current Fig 1 is only reproduction of the existing paper.
- In the last fourth line in page 1 (Please add line numbers for review if possible. It makes reviewing very tricky.), authors substitute the learning rate $\alpha$ to its upper bound for contraction factor, while don't substitute it for the noise floor term. Then it concludes both terms contain $L/c$. This inconsistency is utilized throughout the paper. It is very questionable. I guess you can use the inequality form w.r.t. condition number to avoid this inconsistency (like contraction factor < condition number * $C_1$ and noise floor < condition number * $C_0$, without substitution).
- Above the equation (3), please avoid using the existing probabilistic term 'conditional variance.'
- SciML experiments contribute very little to the paper's theoretical argument. In current status, it just looks like running multiple SGD variants on SciML tasks and check the performance. To relate these results to the paper's content, it would be better to compute (at least estimate) constants ($K, L, \mu$ ...) used in the paper for each environment and preconditioner, to check whether the proposed theory holds empirically.

#### Missing References
- There are several works explaining the benefit of preconditioning compared to pure SGD. Please add them as references.
  - [1] "When does preconditioning help or hurt generalization?" ICLR 2020
  - [2] "On the parameterization of second-order optimization effective towards the infinite width." ICLR 2024
  - [3] "On the concurrence of layer-wise preconditioning methods and provable feature learning." ICML 2025

**Audience:**

Yes

**Audience Explanation:**

Sure, many optimization theory people gonna be interested.

**Broader Impact Concerns:**

I don't think there are ethical concerns.

**Claims And Evidence:**

Yes

**Claims Explanation:**

It seems the arguments are theoretically proven but experiment section seems not.
See the last weakness point.

**Requested Changes:**

- Overall presentation should be improved, especially the introduction section including Fig 1. Currently, it is very difficult to capture what paper want to present, what is the paper's contribution, and novelty. Avoid using undefined (and very contextual) terms from the first page.
- Try to support the theoretical points with experiments. Currently, SciML experiments seem very isolated from the previous sections.

---

### Review · Reviewer_nJHy · 2025-12-22

**Summary Of Contributions:**

This paper introduces a theoretical framework for analyzing preconditioned SGD during the late stages of training, specifically addressing the stagnation caused by anisotropic curvature and gradient noise. The paper derives new convergence bounds in the geometry induced by a preconditioner matrix M, demonstrating that the convergence rate is governed by the effective condition number while the asymptotic noise floor is determined by the product of this condition number and the preconditioned noise level.

- The paper proves that for strongly convex objectives using a fixed learning rate and a preconditioner M, SGD converges linearly down to a specific noise floor. They show that this noise floor is determined by the product of the effective condition number and the preconditioned noise level.

- For non-convex problems, the paper provides guarantees within a well-behaved basin around a local minimizer. They derive a lower bound on the probability that the optimizer remains inside this basin (basin stability). They also demonstrate geometric convergence to a noise floor in this setting.

- The analysis lead to a unified design principle for preconditioners: one should choose the matrix $M$ to simultaneously improve local conditioning and attenuate the gradient noise. This framework is shown to encompass both diagonal/adaptive methods and curvature-aware preconditioners.

- The paper  validates the theoretical findings on a diagnostic quadratic model and different scientific machine learning benchmarks, demonstrating that the predicted behavior (faster rate and lower floor) holds in practice.

**Audience:**

Yes

**Audience Explanation:**

Yes, the findings of this paper are probably of significant interest to TMLR's audience.

- Optimization Researchers: The theoretical decoupling of the convergence rate from the asymptotic noise floor offers a nuanced perspective on late-stage training dynamics.

- Scientific machine learning community: The paper explicitly addresses constraints of SciML, where achieving a very low training loss is not merely a matter of accuracy but of physical fidelity and stability (e.g., satisfying PDE residuals).

- Users of second-order methods: The paper provides a clear design principle for preconditioners: choosing $M$ to jointly improve local conditioning while attenuating noise. This offers practical insight into why curvature-aware methods and adaptive methods often outperform vanilla gradient descents in asymptotic regimes.

**Broader Impact Concerns:**

No broader impact concerns.

**Claims And Evidence:**

Yes

**Claims Explanation:**

This is a theoretical paper, that is validated emppirically. It contains mainly two claims:

**1.  Preconditioning governs both convergence rate and noise floor**

- The paper claims that the convergence rate is determined by the effective condition number in the $M$-metric, while the noise floor is determined by the product of this condition number and the preconditioned noise level. Theorem 3.2 and Theorem 3.4 are the main pieces supporting this claim.

- The paper presents empirical evidence by isolating these variables. By artificially deflating specific eigenvalues of the Hessian, they experimentally verify that improving conditioning (deflating top eigenvalues) lowers the rate, while reducing the trace of the covariance (deflating noise directions) lowers the floor (Figure 2).

**2. Preconditioning improves basin stability**

- The paper derives a probabilistic lower bound for basin stability, showing it depends on the distance to the basin boundary and the local quadratic growth constant. This is done in Theorem 3.4 & 3.5.
- The stability claim is supported by the consistent success of curvature-aware methods in the SciML benchmarks (Sec~4.2). In tasks like the Green's function learning, where the landscape is highly multiscale and stiffness-dominated , standard SGD fails to converge to the low-loss solution, whereas preconditioned methods stably descend to the noise floor.

*Note. I didn't fully check all the mathematical deductions, so my analysis can be slightly superficial.*

**Requested Changes:**

In Assumption 8 there seems to be an apparent contradiction to the experimental Noise models.
- Assumption 8 says that if an iterate is in the basin $N_r$, the next step almost surely remains within $N_{r+}$ . This seems to imply bounded gradient noise which is not compatible to Gaussian noise. Maybe you should relax this assumption to allow a small failure probability $\epsilon$? Can the authors comment on this?

---

### Review · Reviewer_NovZ · 2026-03-23

**Summary Of Contributions:**

This paper provides a theoretical framework for understanding how preconditioning affects the late-stage behavior of Stochastic Gradient Descent (SGD). The authors analyze preconditioned SGD in the geometry induced by a symmetric positive definite (SPD) matrix M, making explicit how both the convergence rate and the stochastic noise floor depend on M.

*Main contributions:*
- Preconditioned SGD in the strongly convex setting: The authors extend classical SGD convergence theory to show that late-stage behavior is controlled by (i) an effective condition number in the M-geometry and (ii) a preconditioned noise level.
- Local nonconvex analysis with basin stability: Under local M-PL and M-smoothness conditions, the paper establishes finite-horizon convergence guarantees within a basin around a minimizer set, along with explicit lower bounds on the probability of remaining in the basin.
- Design criteria: The theory yields a practical principle—choose M to improve local conditioning while attenuating noise in the M^{-1}-norm.
- Experimental validation: The authors validate their framework on a diagnostic quadratic model (where constants admit closed forms) and three SciML benchmarks (Franke surface regression, Poisson PINN, Green's function learning).

*Key strengths:*
- Clean theoretical framework connecting preconditioner choice to both convergence rate and noise floor
- The basin-stability analysis is novel and practically relevant
- Experiments demonstrate the predicted rate-floor behavior

*Key weaknesses:*
- Limited guidance on how to efficiently compute good preconditioners in practice
- Some assumptions (e.g., Assumption 8 on controlled overshoot) may be difficult to verify
- Limited scale of experimetns

**Additional Comments:**

This is a solid theoretical contribution that provides useful insights into preconditioner design for SGD.

**Audience:**

Yes

**Audience Explanation:**

This paper addresses a fundamental question in optimization: how does preconditioning affect SGD convergence? This question might be of interest of optimization researcher as well as deep learning partitioners.

**Broader Impact Concerns:**

No concern related to this work

**Claims And Evidence:**

Yes

**Claims Explanation:**

This paper is motly theoretical. It provides rigorous proofs for all claims, with details in the appendix. The mathematical development is appear to be sound, it follows standard techniques in optimization theory, appropriately adapted to the M-induced geometry.

Paper also provide experimental evidences to support the theoretical predicitions:
-  Quadratic function: The controlled setting allows direct confirmation of the predicted noise-floor scaling.
- SciML experiments : The two-phase protocol (Adam warm-start followed by Phase II comparison) appropriately isolates late-stage behavior. The results consistently show CG-GGN outperforming other methods, which aligns with the theory that GGN-based preconditioners better align with gradient covariance for squared-residual losses.

**Requested Changes:**

- Clarify what is relative loss in the figures 1,2, 3, 4.
- The paper only focus on training loss, and do not investigate the effect on preconditioning on generalization performance. It would be nice to make it more explicit in the paper and discuss how preconditioning might impact generalization.
- Discuss connection to natural gradient: The paper mentions the Fisher Information Matrix but doesn't deeply connect to the natural gradient literature (Amari's work).
- The experiments use relatively small networks . It would strengthen the paper to discuss how the preconditioned methods scale to larger models or provide guidance on when the computational overhead is worth.
- The two-phase experimental protocol is sensible but raises questions about the interaction between Phase I (Adam) and Phase II methods. Does the basin reached by Adam affect which method performs best in Phase II?

---

### Decision · Action_Editor_3pEb · 2026-05-20

**Recommendation:** Accept as is

**Audience:**

Yes

**Audience Explanation:**

The results presented in the paper may be of interest to researchers in stochastic optimization as well as practitioners seeking to better understand the dynamics of preconditioned SGD and to leverage the proposed framework for designing more effective preconditioners.

**Claims And Evidence:**

Yes

**Claims Explanation:**

The paper investigates preconditioned SGD and develops a theoretical framework to characterize the impact of preconditioning on the convergence behavior of SGD across different classes of objective functions. The primary emphasis is on the last stage of SGD, analyzing how both convergence properties and stochastic noise are influenced by the choice of preconditioning matrix. The paper also presents experimental results that support and validate the theoretical analysis.